# Semantic embeddings reveal and address taxonomic incommensurability in psychological measurement

**Dirk U. Wulff** [1,2] ✉ **& Rui Mata** [2]

Taxonomic incommensurability denotes the difficulty in comparing scientific theories due to different uses of concepts and operationalizations. To tackle this problem in psychology, here we use language models to obtain semantic embeddings representing psychometric items, scales and construct labels in a vector space. This approach allows us to analyse different datasets (for example, the International Personality Item Pool) spanning thousands of items and hundreds of scales and constructs and show that embeddings can be used to predict empirical relations between measures, automatically detect taxonomic fallacies and suggest more parsimonious taxonomies. These findings suggest that semantic embeddings constitute a powerful tool for tackling taxonomic incommensurability in the psychological sciences.

Taxonomic incommensurability, the idea that various scientific theories or paradigms are often incomparable due to their distinct mappings between theoretical concepts and measures or empirical results[1,2], poses considerable challenges to all sciences. For the social and behavioural sciences, such as psychology, the lack of a clear mapping between constructs and operationalizations results in the difficulty of selecting appropriate measures for describing, predicting and changing behaviour (for example, ref. 3). Consequently, addressing taxonomic incommensurability is vital for any discipline interested in understanding and improving human health, wealth and well-being.

Recent years have seen numerous calls for conceptual clarification in psychology (for example, refs. 4,5); specifically, the need to address 'conceptual clutter'[6], 'concept creep'[7] or 'jingle–jangle fallacies'[8–11]. These terms suggest that psychology is plagued by the proliferation of measures that have been given similar labels yet may capture different constructs (jingle fallacy), whereas other measures have received different labels yet capture the same construct (jangle fallacy). This problem represents a form of taxonomic incommensurability and has long been recognized as stemming from the lack of a common nomological network[12] or ontology (for example, ref. 13), that is, an agreed-upon conceptual lexicon consisting of concepts and their observable manifestations that can be used to characterize human psychology.

Previous work has shown that natural language processing methods can be used to help clarify the relations between constructs and their measures (for example, refs. 11,14,15). Crucially, some approaches have been shown to identify jingle–jangle fallacies in an automated fashion across a large swathe of constructs[11]. However, these previous efforts have used techniques, such as latent semantic analysis, that have now been superseded by more powerful language models that promise to capture several aspects of human psychology even better (for example, refs. 16–18). A thorough comparison of different large language models, and an approach to detect jingle–jangle fallacies using these models, is lacking. Moreover, previous efforts have only considered gradual differences in the relatedness of measures but not fully the interconnections between constructs and their operationalizations, which is crucial for a full conceptual understanding of jingle–jangle fallacies, improving the mapping between measures and constructs, and reducing jingle–jangle fallacies through an improved taxonomy.

Our work aims to help tackle taxonomic incommensurability by introducing an approach that uses large language models to create a quantitative depiction of psychological constructs and their linguistic operationalizations (for a tutorial, see ref. 18). This method relies on item, scale and label embeddings—vector space representations of psychometric items, scales and their respective construct labels—to

[1]Center for Adaptive Rationality, Max Planck Institute for Human Development, Berlin, Germany. [2]Center for Cognitive & Decision Sciences, University of Basel, Basel, Switzerland. ✉e-mail: wulff@mpib-berlin.mpg.de

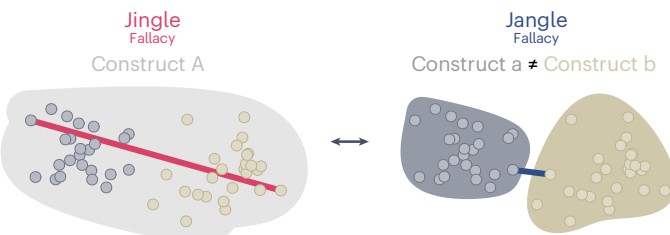

**Fig. 1 | Jingle–jangle fallacies.** The figure illustrates jingle–jangle fallacies and the associated trade-off between minimizing jingle and jangle fallacies. Each point represents a scale, with its placement being determined by its similarity in the semantic space to other scales; convex-hull polygons are drawn around the points to represent the construct labels shared by a group of scales. The group of scales can be assigned labels either manually or in an automated fashion. Left: a labelling process may avoid jangle fallacies by giving all related scales the same construct label (construct A). In contrast, this may cause some unrelated scales that are distant in the similarity space to have the same label, thus creating one (or more) jingle fallacies (that is, red line). Right: a labelling process may avoid jingle fallacies by creating two (or more) semantically distinct labels (construct a and construct b) that maximize within-construct similarity for two groups of scales. However, this approach may lead to cases in which very similar scales have been assigned the two semantically distinct labels, thus creating a jangle fallacy (blue line).

map the semantic relations between an extensive array of psychological measures and the putative constructs they represent (see 'Terminology' in Methods for more detailed explanations of the terms used in this article). In our work, we make the following contributions: first, we validate different embeddings, including embeddings obtained from a model specifically fine-tuned for our purposes, by testing how they capture known empirical relations between different psychological measures. Second, we introduce a method to perform automated detection of jingle–jangle fallacies in existing taxonomies by relying on the best-performing embedding. Third, we introduce a clustering and relabelling approach to deal with the inherent trade-off involved in minimizing jingle and jangle fallacies (Fig. 1). Overall, our approach tackles taxonomic incommensurability by boosting conceptual clarity and providing a more parsimonious taxonomy of constructs in psychology.

In what follows, we provide an overview of our approach using the data from the publicly available International Personality Item Pool (IPIP; http://ipip.ori.org)[19]. Our approach relied on obtaining item, scale and label similarities from embeddings following a number of steps illustrated in Fig. 2. Specifically, we included items comprising 459 scales and capturing 277 constructs encapsulated in 254 unique single- and multi-construct labels. First, we obtained item embeddings for the 4,452 psychological items in IPIP (Fig. 2a). We obtained embeddings using different models but throughout this article we present findings based on a fine-tuned model, which we make publicly available and which proved the most powerful in the validation efforts described below. This fine-tuned model is based on the MPNet pre-trained model[20], which builds on the bidirectional encoder representations from transformers (BERT) architecture and is one of the best available sentence-BERT models[21]. The model was fine-tuned using 200,000 pairs of personality items from several open datasets and is publicly available on Hugging Face (dwulff/mpnet-personality). We provide a detailed description and systematic comparisons with other embeddings in the Supplementary Information (Supplementary Figs. 1–5)[15,20,22–24]. Second, we obtained scale embeddings by averaging the embeddings for the items corresponding to each of the 459 scales in IPIP (Fig. 2b). Third, we obtained label embeddings for 277 distinct labels produced by IPIP's construct labels. Specifically, from the original 254 IPIP labels, we used those consisting of unique words and further split multi-construct labels into single ones to potentially

capture each construct individually, resulting in 277 distinct labels. Fourth, using the item, scale and label embeddings, we derived similarities between all pairs of items, scales and labels by computing the cosine similarity between the respective vectors in the embedding space (Fig. 2d–f). Finally, we matched scales and labels in this space (Fig. 2g). Once equipped with embeddings and respective similarities for items, scales and labels, we were in a position to ask whether this information is able to capture meaningful relations between measures and address the problem of jingle–jangle fallacies.

## Results

Our results are divided into three sections. First, we provide a demonstration of the ability of embeddings to recover known relations between psychological measures, such as the patterns of convergent and divergent validity obtained from self-reports. Second, we put embeddings to use in an effort to address conceptual clutter in psychology by detecting jingle–jangle fallacies in an automated fashion. Third, we introduce an approach to providing a more parsimonious mapping between constructs and extant psychological measures.

### Validation of embeddings

We carried out four analyses aimed at addressing the power of different embeddings for recovering known properties of psychological measures, including their internal consistency, structural fidelity, patterns of convergent and divergent validity, and the match between scale content and their labels. In what follows, we provide our results for a fine-tuned model that performed best overall in validation. See the Supplementary Information for details on the fine-tuning procedure and results of all models. We provide values for in-sample and out-of-sample performance. The former reflects how well the model captures empirical relations concerning items it had access to during training, whereas the latter reflects the model's generalization performance.

First, we validated embeddings by assessing whether scale similarities obtained from embeddings are able to predict a scale's empirically observed internal consistency; that is, Cronbach's $\alpha$ (ref. 25), a measure typically used to capture the average empirical correlation between items from the same scale. A strong positive correlation would suggest that the semantic similarity between items can be used to quantify the observed empirical correlation from self-reports, which represents a minimum requirement for a conceptual representation based on embeddings if this is to inform us about the structure of psychological measures more generally. To this end, we estimated the within-scale similarity and then used the Spearman–Brown prediction formula to derive estimates of the scale's Cronbach's $\alpha$. As illustrated in Fig. 3a, we observed a strong Pearson correlation between the observed and predicted internal consistencies of $r = 0.75$ (in sample) and $r = 0.61$ (out of sample) for the 449 scales for which empirical scores were available in IPIP.

Second, we compared to what extent item similarities obtained from item embeddings capture scales' structural fidelity, that is, a metric quantifying whether a set of items is most strongly related to other items from their scale of origin relative to items stemming from other scales. Showing some degree of structural fidelity would suggest that embeddings allow matching between items and scales. Specifically, we computed a $z$-score for each scale that reflects the similarity of items within the same scale relative to the similarity of items of different scales within an inventory. We regard the structure of a scale as fully recovered if $z > 2$ and partially recovered if $z > 1$. These criteria roughly match the lower bound obtained from following the same procedure using the observed empirical correlations. We evaluate structural fidelity for the 25 inventories with at least 3 scales that, on average, had 12.4 scales. The results show that 81% (in sample) and 79% (out of sample) of all 443 scales were recovered and 95% (in sample and out of sample) were partially recovered (Fig. 3b), suggesting that embeddings capture the structure of a large number of psychological inventories.

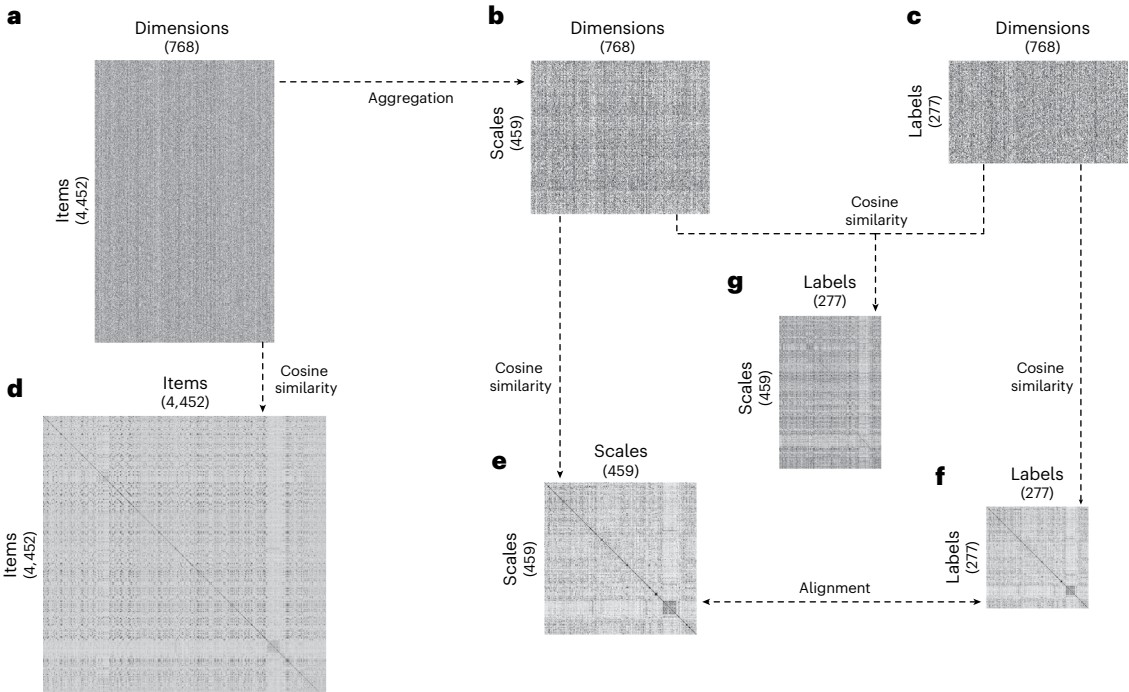

**Fig. 2 | Overview of the analytic approach using item, scale and label embeddings. a–c,** The embeddings concerning personality items (**a**), scales (**b**) and labels (**c**) from the IPIP (http://ipip.ori.org)[19] are shown. The embeddings are obtained using our fine-tuned embedding model (Supplementary Information), and each column represents 1 of 768 dimensions of the embeddings model. In **a**, each row represents 1 of 4,452 IPIP items; in **b**, each row corresponds to 1 of 459 scale embeddings obtained by averaging the item embeddings associated with each respective scale in IPIP; in **c**, each row refers to 1 of 277 labels present in IPIP. **d–f,** Shown are all pairings of items (**d**), scales (**e**) and labels (**f**), respectively, obtained by computing the cosine similarity between the embeddings shown in **a–c. g,** The similarity between labels and scales using the cosine similarity between the two, as defined by their respective embeddings.

Moreover, we specifically compared the predicted structural fidelity to that obtained from five publicly available datasets on the IPIP-NEO-300, HEXACO, BIG5, FFM and 16PF personality inventories. These inventories are composed of 5 (BIG5 and FFM) to 30 (NEO) scales (Fig. 3b; see Supplementary Information for details). For the empirical data, we evaluated structural fidelity in an analogue fashion by first determining the average correlation of items within and between scales and then calculating the ratio of the within correlation to the strongest correlation for a given scale. Our analysis suggests that structural fidelity is similar between the empirical data and the predictions of the model. We observed a strong correlation between the predicted and empirical structural fidelity, ranging from $r = 0.76$ (16PF) to $r = 0.85$ (HEXACO) for the out-of-sample model, further underpinning the fact that embeddings can largely capture the psychometric structure of psychological scales and inventories.

Third, we assessed more generally the ability of embeddings to make predictions about convergent and divergent validity by using item and scale similarities to make predictions about the observed correlations between items and scales across various personality constructs. Specifically, we compared the predicted correlation between items and scales based on the sentence embeddings with the empirical correlations found in different datasets, excluding self-correlations. We compared the predictions to the empirical absolute correlations, reflecting the relatedness between items independent of direction. At the level of items, we observed extremely low mean absolute errors (MAEs) between MAE = 0.026 (FFM) and MAE = 0.029 (16PF) in sample and higher but still low errors between MAE = 0.07 (BIG5) and MAE = 0.065 (HEXACO) out of sample. These imply correlations of, on average, $r = 0.96$ (in sample) and $r = 0.64$ (out of sample). At the level of scales, we observed extremely low MAEs between MAE = 0.037 (BIG5) and MAE = 0.047 (16PF) in sample and somewhat higher but still low between MAE = 0.12 (NEO) and MAE = 0.14 (HEXACO) out of sample.

These imply correlations of, on average, $r = 0.92$ (in sample) and $r = 0.63$ (out of sample). Overall, these results show that embeddings accurately capture the unsigned empirical correlations between items and scales.

Fourth, to identify jingle–jangle fallacies, it is important to have not only powerful item and scale embeddings but also a good mapping between these and the constructs themselves, for example, as operationalized by label embeddings. In our validation strategy, we generated several types of label embeddings, which we reasoned could have differential strengths and weaknesses, and evaluated their relative alignment to scale content as operationalized by the scale embedding, using a $z$-score quantifying the similarity between scales and their associated label relative to all other labels. Specifically, we considered five types of label embeddings: the construct labels present in IPIP, contextualized labels (that is, a version of the construct label produced by placing the construct label in the sentence: 'The personality construct [LABEL].'), the American Psychological Association (APA) construct definitions, a manually curated version of the APA construct definitions and several variations of labels generated by GPT-4 (for example, different prompts). We provide a more detailed rationale for these different types in Methods but, in summary, we reasoned that although IPIP labels provide wide coverage, they are typically short and some are used in everyday language without explicit reference to the personality construct (for example, 'warmth' and 'organization'), which can present a challenge to obtaining appropriate, construct-specific embeddings. In turn, other approaches that may provide richer embeddings that are closer to the human expert definition (for example, APA definitions) do not provide full coverage of all IPIP constructs or may partly result from hallucinations (for example, GPT). Consequently, an empirical test of the alignment between different scale and label embeddings is in order. In practice, we computed an alignment score for each scale and label embedding using cosine similarity and averaged the alignment scores across scales to obtain an overall estimate of how well a given

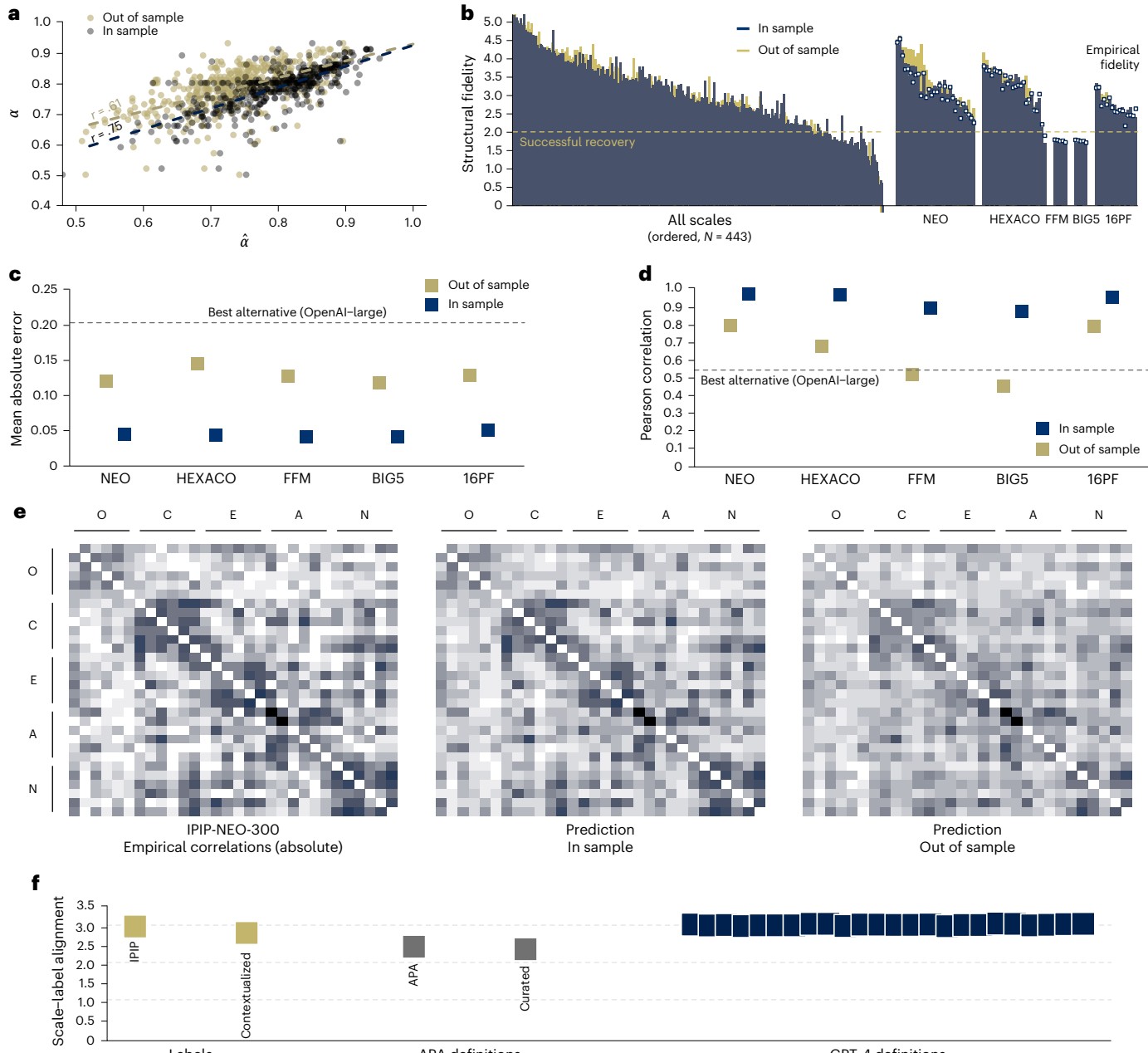

**Fig. 3 | Validation of embeddings. a**, The correlation between the empirical ($\alpha$) and the predicted $\hat{\alpha}$ internal consistency concerning in-sample (blue) and out-of-sample (yellow) predictions from our fine-tuned model for 449 scales. **b**, The structural fidelity (defined as the z-score) is validated, contrasting the similarity of items within a given scale to items of others scales within the same inventory for all inventories with at least 3 scales (N = 433). In addition, it contrasts the structural fidelity that embeddings achieve for five empirical datasets (NEO, HEXACO, FFM, BIG5 and 16PF) relative to the observed structural fidelity (represented as points) obtained from large-scale empirical assessment. **c,d**, The mean error (**c**) and correlation (**d**) between the predicted and observed scale correlations for the same five datasets is shown. **e**, A visual comparison of the observed correlation among the 30 scales of the IPIP-NEO-300 in 1 empirical dataset and the in-sample and out-of-sample predictions based on the embeddings. The correlations are ordered along to the personality factors Openness (O), Conscientiousness (C), Extraversion (E), Agreeableness (A) and Neuroticism (N). **f**, Illustration of our construct label evaluation. The points show the average alignment score as defined in the text for embeddings of the labels of distinct constructs present in IPIP, contextualized labels, APA definitions, curated APA definitions and definitions generated by GPT-4.

instantiation of label embeddings is aligned with the corresponding scale embeddings. Figure 3f shows the results for the different types of label embeddings considered. The original IPIP labels showed a high alignment, implying that, on average, the similarities between scale and matching label embeddings were over 2 s.d. higher than those between scale and non-matching construct embeddings. Alignment was not improved using the contextualized labels. The original and manually curated APA definitions also showed lower alignment, which may be partly explained by only a portion of scales being considered. Yet, even when constraining to the available data, the labels and the GPT-4 definitions showed substantially higher alignment. With both labels and GPT-4 embeddings showing comparable alignment, we decided to rely directly on the labels for the analyses presented below. The main motivation for this decision was to remain closer to the original labels while shielding our analysis from possible generative hallucinations of the GPT-4 model.

Overall, these results show that embeddings, and particularly our fine-tuned model, can capture a number of central characteristics of psychological measures and their interrelations. Scale embeddings show performance levels that are higher than previous results obtained using small(er) language models[15,16] and on par with fine-tuning approaches developed concurrently[26]. Further, label embeddings seem to provide considerable alignment between scale content and associated constructs, as defined by scale labels or construct definitions. In what follows, we capitalize on the power of our fine-tuned embedding model to provide an overview of the similarity between these measures and their respective labels to identify and minimize jingle–jangle fallacies in psychology.

## Automated jingle–jangle detection

We put embeddings to use by automating the detection of jingle–jangle fallacies, that is, making explicit the relative mismatch between scale content and their respective labels. To this end, we leveraged the fact that both scale content and scale labels are amenable to an analysis of semantic similarity using embeddings (Fig. 2g). Specifically, we identified jingle–jangle fallacies by identifying criteria that quantify high similarity in construct labels associated with low similarity in the associated scales—jingle fallacy—or, alternatively, low similarity in construct labels associated with high similarity in the associated scales—jangle fallacy.

One open issue with using similarity metrics is that a criterion must be chosen to define whether a difference is large (or small) enough to warrant considering two things as different (or equivalent). Given there is no single, straightforward answer to this problem, we considered different criteria in our work. Crucially, although the absolute number of fallacies varies as a function of the specific criterion used (Supplementary Information), the relative proportion of jingle and jangle fallacies is similar across different criteria and so can be thought of as a characteristic of the current mapping between constructs and measures.

In what follows, we report results based on an arbitrary but principled heuristic approach informed by previous work in personality psychology. To begin, we reasoned that there should be a range of expected numbers of distinct constructs within the IPIP data and that it would be reasonable to assume that this range is somewhere between a lower bound of 5 and an upper bound of 100 constructs. There is considerable work in personality psychology that assumes that personality structure involves at least five constructs (that is, big five), consequently, that could represent a plausible lower bound. If this were true, a logical consequence assuming equal distribution of measures across constructs is that 1 of 5 of measures should belong to the same construct and, by extension, 4 of 5 to different constructs. This implies that measure or construct pairs in the lower 80% of the similarity distribution belong to distinct measures and constructs, leading us to define the similarity quantile belonging to 0.8, corresponding specifically in our data to cosine values of 0.37 (measures) and 0.25 (labels), as the threshold for identifying distinct measures and constructs. In turn, concerning the upper bound, we reasoned that the true number of distinct constructs represented in IPIP is, or at least should be, lower than the 254 multi-construct or 277 single-construct labels because of the frequent criticisms of an excessive proliferation of constructs discussed above. If the true number of constructs was identical to the number of labels, then no pair of distinct construct labels (for example, mistrust and distrust, or understanding and comprehension) would represent a jingle fallacy. In what follows, we adopted an upper bound that assumes the existence of 100 constructs, which implies 1 of 100 of the similarity distribution belongs to identical measures and constructs. This leads to defining the corresponding similarity quantile, namely, cosine values of 0.64 (measures) and 0.50 (labels), as the threshold for identifying identical measures and constructs.

Applying this heuristic criterion, we identified a total of 504 fallacy candidates, specifically, 340 jingle and 164 jangle fallacies.

Figure 4 shows these comparisons, highlighting a subset of cases in which the two fallacies diverge. It can be seen that there exist numerous cases in which the similarity of a pair of scales' labels far exceeds the similarity of the pair's content—a jingle fallacy. In turn, there are numerous scale pairs with highly dissimilar labels whose scale similarities are strikingly high—a jangle fallacy. A more thorough understanding of these candidate fallacies requires an assessment of the specific items and their respective labels, so we provide a few examples of jingle and jangle fallacies in the two paragraphs below.

A minority (5.9%) of the candidate jingle fallacies identified by our method concerns scales of the same constructs in different inventories. For instance, the 'orderliness' scales in the 6FPQ and HPI-HIC only show a scale similarity of cosine = 0.3, which can be attributed to the former concerning cleanliness (example item, 'leave a mess in my room') and the latter tradition ('dislike routine'). The vast majority of candidate jingle fallacies involves different but highly similar constructs. Most concern the constructs 'organization' (35, 10.3%), 'humour/playfulness' (24, 7.1%), 'toughness' (23, 6.8%), 'industriousness/perseverance/persistence' (22, 6.5%), 'friendliness' (20, 5.9%) and 'adventurousness' (19, 5.6%). Overall, 129 of 277 constructs are part of at least 1 candidate jingle fallacy. For instance, the constructs organization and orderliness are highly similar, whereas corresponding scales in the AB5C and HPI-HIC differ for the same cleanliness versus tradition distinction mentioned above. Similarly, 'friendliness' and 'amiability' are highly similar (cosine = 0.66), whereas the corresponding scales in the TCI and CPI are rather different (cosine = 0.3), with the former capturing irritability ('am often in a bad mood') and the latter sociability ('warm up quickly to others').

The number of candidate jangle fallacies identified by our method involve scales from 80 different constructs. Many candidate jangle fallacies concern the constructs 'neuroticism' (40, 38.4%), 'emotional stability' (23, 22.1%), 'happiness' (20, 19.2%), 'security' (17, 16.3%) and 'anger' (15, 14.4%). The majority of candidate jangle fallacies can be explained at least partially by shared items. Across all jangle fallacy pairs, scales shared an average of 3.8 items of an average of 9.7 possible item overlaps. For instance, in the candidate jangle fallacy concerning the 'conscientiousness' scale in the BIG5 inventory and the organization scale in the JPI inventory, seven items are shared (for example, 'follow a schedule' or 'make plans and stick to them'). We note, however, that item overlap does not fully explain the existence of jangle fallacies. The number of overlapping items is imperfectly related to the cosine similarity between scales ($r$ = 0.59). Moreover, there are 75 candidate jangle fallacies with 1–3 overlaps, which do not automatically result in a fallacy, given that there are 3,785 scale pairs with 1–3 overlaps that were not identified as a fallacy. There were also seven candidate jangle fallacies with zero item overlap. One example of a low-overlap fallacy is happiness in the AB5C inventory and 'forcefulness' in the CPI, with both scales capturing similar aspects of expressing confidence (for AB5C, 'feel comfortable with myself'; for CPI, 'am very pleased with myself') or scepticism (for AB5C, 'am filled with doubts about things'; for CPI, 'am afraid that I will do the wrong thing') about oneself.

To summarize, our analysis suggests that it is possible to automate jingle–jangle detection using embeddings and that both fallacy types can be detected in existing taxonomies, such as IPIP. In what follows, we address possible ways of leveraging automated jingle–jangle detection to minimize these fallacies and the associated trade-off.

## Minimizing jingle–jangle fallacies to increase conceptual clarity and parsimony in psychology

The ability of embeddings to place both scales and labels in the same representational space offers an opportunity to reorganize and relabel measures with the goal of minimizing jingle–jangle fallacies while potentially achieving greater parsimony in the number of constructs proposed. We investigated this possibility by developing a procedure consisting of the following three steps. First, we used clustering

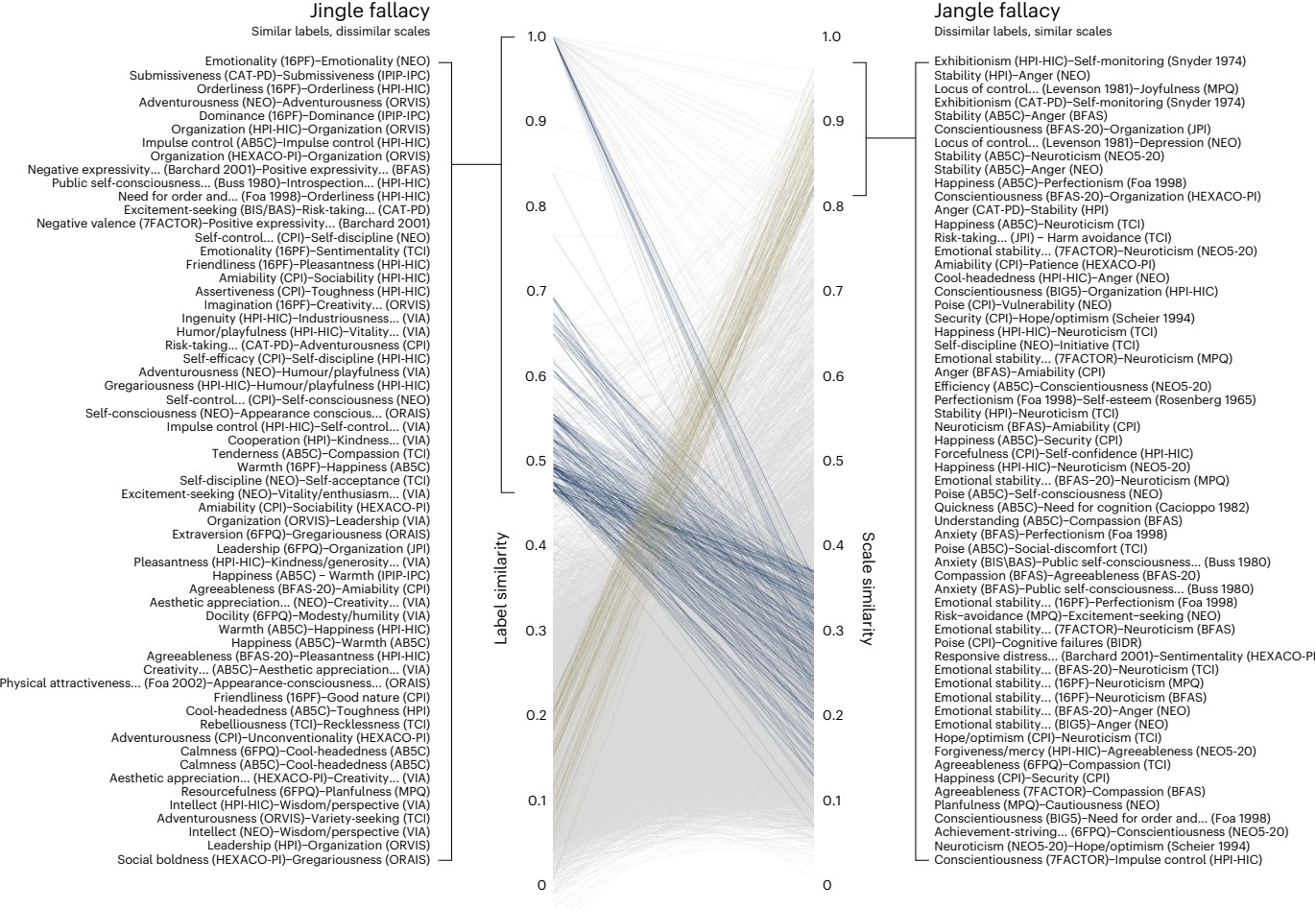

**Fig. 4 | Automated jingle–jangle detection.** The figure contrasts the similarity of scales' labels (label similarity) and their content (scale similarity) for all pairs of the 459 scales analysed. Blue and beige lines and corresponding text highlight those pairs that are either high in label similarity but low in scale similarity (blue) or vice versa (beige). The labels to the right and left of the line plot show a random selection of fallacy pairs.

algorithms to identify clusters of scales based on their semantic similarity (Fig. 2e) and allowing for a variety of possible numbers of clusters (for example, 1–277). The rationale for using a number of clustering algorithms is that these emphasize different criteria when producing clustering solutions so it is important to test the generality of conclusions from any single one[27]. Second, for each clustering algorithm and solution (that is, number of clusters), we used a maximum bipartite matching algorithm[28] to assign construct labels to measures such that the semantic similarity between measures and labels (Fig. 2g) was maximized. Third, we evaluated for each solution the number of jingle and jangle fallacies using the method described in the previous section for each of the new scale–label mappings and used this information to evaluate the overall reduction in jingle and jangle fallacies that were obtained as a function of the number of clusters.

Figure 5a shows the result of this approach for the clustering solutions produced by one specific algorithm: hierarchical clustering using Ward linkage. The figure reveals a general reduction in the number of fallacies as the number of clusters grows larger but also a trade-off between minimizing jingle and jangle fallacies. In this case, the total number of fallacies is optimized by a solution with 150 clusters. This solution produces a total of 154 fallacies, including 136 jingle and 18 jangle fallacies, which is less than 1/3 of the 540 fallacies produced by the IPIP label assignments. However, there exist more parsimonious solutions that propose fewer clusters while incurring only a few additional fallacies. One such solution uses 68 clusters and produces

228 fallacies (215 jingle and 13 jangle fallacies)—that is, only a few dozen additional fallacies relative to the optimal solution—while proposing less than half the number of constructs (68 versus 150, 45%). Our evaluation of four other clustering algorithms revealed similar patterns, including the production of fallacy-reduced parsimonious solutions, which we defined as best solutions under 100 clusters, which we assumed for 1 of the similarity thresholds. Specifically, as can be seen in Fig. 5b, we found that most algorithms produce a smaller or comparable number of fallacies proposing substantially fewer clusters, with the exception of single hierarchical clustering, which can be explained by its strong focus on minimizing the minimum pairwise distance, leading to minimizing primarily jangle fallacies. We present full results for all clustering algorithms and solutions in the Supplementary Information. Overall, these results suggest that our approach can reliably produce a smaller number of fallacies than IPIP, with many possible solutions being parsimonious in the sense of identifying substantially fewer clusters and, in effect, proposing a more parsimonious representation of constructs in psychology.

Figure 5c provides a visual illustration of a possible mapping between measures and labels produced by the parsimonious solution of the hierarchical clustering with Ward linkage, which can be considered the best compromise between the number of fallacies and the number of clusters across all solutions found. The mapping assigns 375 of 459 measures to 1 of the top 5 most semantically similar labels (solid lines), implying that 84 measures are assigned a less proximate label

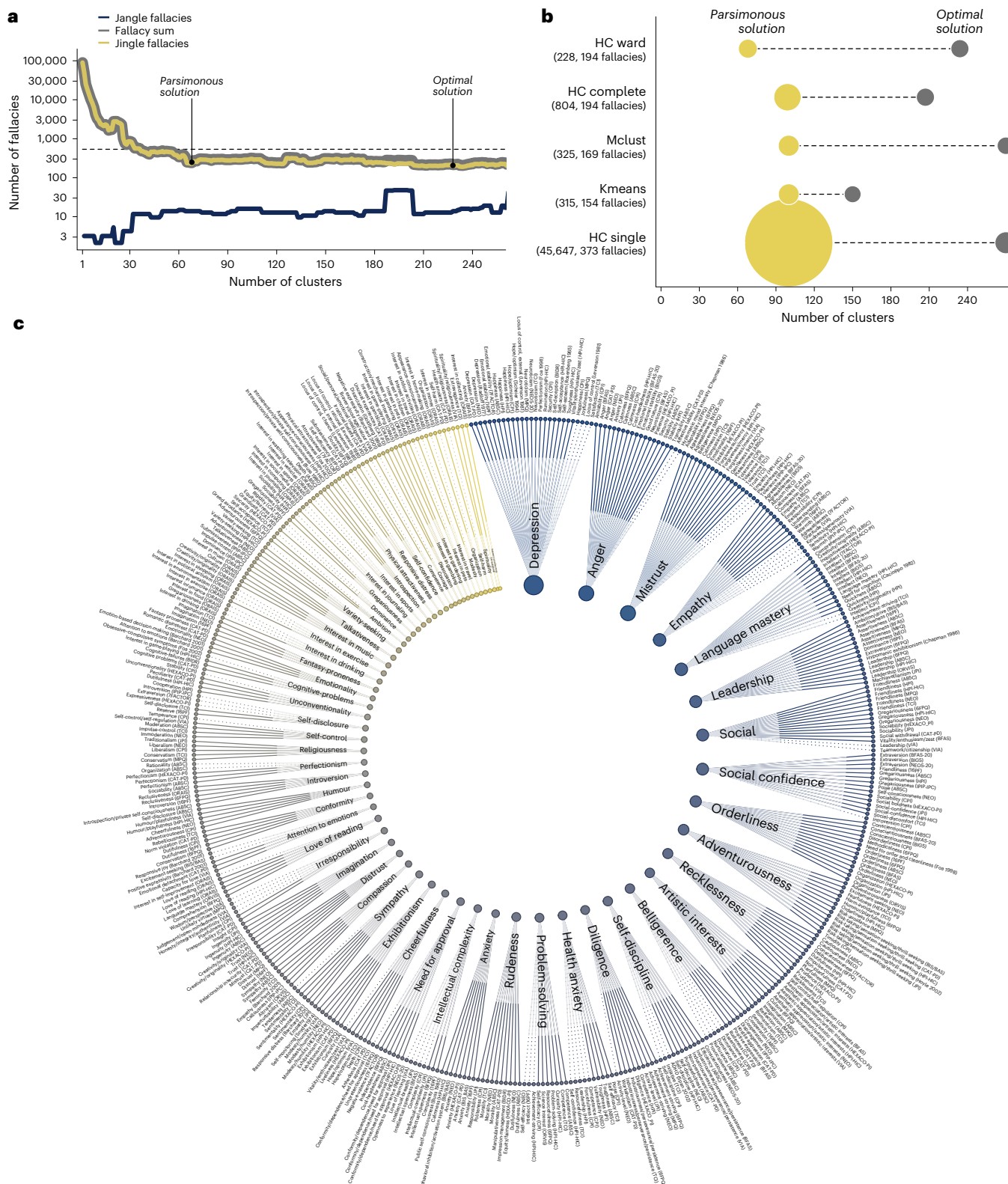

**Fig. 5 | Minimizing jingle–jangle fallacies. a**, The number of fallacies as a function of a number of clusters proposed by a specific clustering algorithm (hierarchical clustering using Ward linkage (HC ward)). The plot shows the trade-off between minimizing jingle and jangle fallacies as a function of the number of clusters and highlights both the most parsimonious solution that minimizes the number of clusters while reducing fallacies and the optimal solution that considers the minimization of fallacies only. The dashed line indicates the number fallacies produced by the original IPIP label assignments. **b**, The number of clusters and fallacies across various clustering algorithms. For each algorithm, the most parsimonious solutions are determined as those

with the fewest fallacies using no more than 100 clusters and constructs (the arbitrary criterion for the maximum number of constructs used in the example). The optimal solutions are those with the fewest fallacies. Kmeans and Mclust refer to *k*-means and Gaussian mixture clustering, respectively. The numbers in parentheses indicate the number of fallacies produced by the parsimonious and optimal solutions, respectively. **c**, A radial plot indicating a possible mapping of 459 scales to 68 labels based on the parsimonious solution shown in **a**. Solid and dashed lines reflect assignments of optimal (top-five rank) and suboptimal similarity, respectively.

with an average rank of 15.3. This is considerably more coherent than the mapping implied by the IPIP labels (despite using the maximum similarity for multi-construct labels), which only assigns 244 measures a top-5 label and otherwise assigns a rank of 47.1. Crucially, this improvement is achieved without multi-construct compounds and using only 68 labels, which is about a quarter of the construct labels implied by IPIP. In sum, our approach used language embeddings to reorganize and relabel existing psychological measures, resulting in a streamlined mapping incurring 58% fewer jingle–jangle fallacies while requiring 75% fewer unique labels.

Our analysis of the IPIP data suggests that our approach is promising but may be limited in scope by focusing on a subset of constructs and measures in psychology. To deal with this limitation, we also conducted a conceptual replication of our approach using an additional dataset (Supplementary Information)[15]. To summarize, the results show the generalizability of our approach in its ability to identify jingle–jangle fallacies and associated trade-offs, whereby eliminating one type of fallacy increases the other, and generate more parsimonious solutions that reduce both jingle–jangle fallacies and the number of constructs proposed.

Overall, although our approach does not offer a full elimination of jingle–jangle fallacies, it suggests that automated methods based on embeddings can render these fallacies more transparent and increase conceptual clarity in psychology.

## Discussion

We heeded recent calls to address taxonomic incommensurability in psychology by using language embeddings obtained from different language models to clarify the relation between constructs and their measures in the psychological sciences. Our work makes three main contributions. First, we compare a number of embeddings, including a fine-tuned model, and identify those that can best recover key structural features of human psychology, including the internal consistency and the convergent validity for a wide range of psychological measures. Second, and crucially, establishing the validity of embeddings warrants us to advance a new approach to addressing conceptual confusion in the psychological sciences. Our approach relied on the fine-tuned model to quantify the similarity between different construct labels and associated measures. This approach presents a way of identifying (mis)alignment between constructs and their operationalizations and determining the existence of jingle–jangle fallacies in an automated, quantitative and reproducible manner. Third, we showed that embeddings can be used not only to identify but also to reduce the number of jingle–jangle fallacies and consolidate the space of constructs in psychology. Crucially, our results suggest that a smaller set of the constructs considered may be sufficient to retain considerable granularity as represented in the semantic space of embeddings.

The methods introduced here have considerable potential to inform future conceptual and measurement work. Conceptually, language embeddings offer a tool to establish relations between extant and future constructs in psychology, which could be important for theory development, particularly in those areas, such as personality, for which many overlapping constructs have been proposed (for example, refs. 29,30). Methodologically, the use of embeddings promises to be helpful in designing new items and scales that maximize fit to well-defined constructs while minimizing overlap with existing measures (for example, refs. 15,16). Further, these methods could be fruitfully applied in a range of research areas that have struggled with jingle–jangle fallacies, such as emotion (for example, ref. 10), work and organizational behaviour (for example, refs. 14,31), learning and pedagogy (for example, ref. 32), or yet broader ontologies that specify more complex relations between theoretical concepts and operationalizations (for example, refs. 3,13). Finally, this approach could be helpful to other disciplines, including economics, political science or

sociology, that also rely on verbal theory and language-based methods to capture a swathe of constructs, attitudes and beliefs.

We note three main limitations of our work. First, we should note that our validation results are based on a limited set of data that may not fully generalize to the psychological literature at large or other sciences. Future work should expand our approach by adopting larger sets of construct definitions and measures both in validation and application efforts. Regarding applications, our results suggest that although our main findings concerning jingle–jangle fallacies and (lack of) parsimony of conceptual spaces generalize across applications as they apply to both personality (that is, IPIP) and broader sets of measures[15], they also suggest important differences to be examined in future work. One crucial step in this effort will involve the careful curation of construct labels and definitions to be used by language models, possibly by expanding existing databases (for example, APA Dictionary of Psychology) or explicitly curating new ones through expert consensus.

Second, we adopt only one form of reducing jingle–jangle fallacies—relabelling of existing scales using a sequential process. Alternatively, future research could consider developing more holistic custom algorithms capable of simultaneously solving the grouping and relabelling problems while minimizing jingle–jangle fallacies. In addition, there are further alternative approaches that could be pursued, such as eliminating scales or changing scale composition to generate a more coherent mapping between constructs and their measurement. One promising alternative is the assignment of items to constructs directly to create new, purer scales of specific constructs. Future empirical work could explore the predictive value of this item-level approach, perhaps by leveraging large-scale data at the item level to compare the predictive validity of synthetically generated scales for predicting relevant real-world criteria[33].

Third, despite the apparent consensus for the need for conceptual clarity in the psychological sciences (for example, refs. 4,5), some researchers have suggested that efforts to reduce the number of concepts may be counterproductive by limiting the consideration set that may be needed, given disparate coexisting goals[34]. We would like to suggest, however, that our approach should not be seen as a substitute for the need for a pluralistic discussion of concepts and the needs they address but as an empirical, quantitative aid to foster debate. Indeed, our approach requires hypothesizing a minimum/maximum number of constructs needed in a specific research area, something that will probably be best established through broad expert consensus. In practice, this may amount to expert meetings involving diverse perspectives or other consensus-building techniques to discuss the utility of different constructs. In addition, it could be helpful to consider changing practices for scale development and associated journal review policy for new constructs and measures that are based on openly searchable semantic spaces that may rely on some of the techniques we adopt here[15].

Overall, despite these limitations, our work shows the potential of language embeddings for the automated detection and minimization of jingle–jangle fallacies and the production of more parsimonious taxonomies of constructs to tackle taxonomic incommensurability.

## Methods

### Terminology

Our contribution focuses on psychometric instruments or measures that consist of collections of linguistic statements that typically describe a state, attitude or disposition (for example, 'I enjoy thinking about things') and that traditional approaches have asked human respondents to endorse or rate on some ordinal scale. Single statements are typically termed 'items' in the psychological literature. The large majority of personality measures uses several such items to capture the same underlying psychological construct, with a collection of two or more related items being typically termed a 'scale'. Measures

consisting of several scales may be used to capture a wide range of constructs, and one such collection is typically termed an 'inventory'. We use the term 'label' to refer to the construct (and associated label) thought to underlie the trait captured by an item, scale or inventory. Finally, we use the term 'embedding' to refer to the representation of linguistic units, such as words or sentences, in a mathematical form, for example, as a real-numbered vector. As we describe below, there are a number of language models that can be used to generate embeddings from linguistic units, with the most recent of these involving billions or trillions of parameters and that, because of their size, are often called large language models. Embeddings can be used to characterize single items, scales, or their labels, and we use the terms 'item embedding', 'scale embedding' and 'label embedding', respectively, for these.

## Personality measures and data

We use several data sources in our work. First, we rely on a large pool of personality measures from the IPIP (http://ipip.ori.org), which currently consists of 4,452 items and 459 scales belonging to 27 multi-scale and 10 single-scale inventories. IPIP was created to refine and improve personality assessment by adapting item and scale labels and using multi-construct labels, for example, risk-taking/ sensation-seeking/thrill-seeking, to capture the tremendous overlap in these constructs. IPIP contains 254 labels, including 24 multi-construct labels, that contain 277 individual construct labels. In addition, we use the IPIP's estimates of the scale's internal consistency (that is, Cronbach's $\alpha$) available for 448 of the 459 scales in our validation analyses.

Second, we obtained definitions of personality constructs from the APA dictionary (https://dictionary.apa.org/) and generated definitions from GPT-4. For details concerning the generation process, see 'Label embeddings'.

Third, we use a number of data sources to conduct fine-tuning and perform validation of the embeddings. We used data from the Open-Source Psychometrics Project, a citizen-science project gathering personality ratings via an online platform (openpsychometrics. org/) that covered 4 inventories: the 16PF inventory, consisting of 162 items and 16 scales; the BIG5 inventory, consisting of 50 items and 5 scales; the FFM inventory, consisting of 50 items and 5 scales; and the HEXACO inventory, consisting of 240 items and 24 scales. We restricted the data to US respondents, resulting in 23,988 (16PF), 19,719 (BIG5), 546,403 (FFM) and 15,017 (HEXACO) respondents. Further, we used data from ref. 35 (www.psycharchives.org/en/item/e42a4531-1daa-4f3d-aef4-58f085c77cd8), which involved a large-scale citizen-science assessment of the NEO inventory consisting of 300 items and 30 scales. We used the data from 212,625 US respondents. Finally, for the purpose of fine-tuning the model, we used data from the Eugene-Springfield Community Sample (dataverse.harvard.edu)[36], including items from the NEO, BIG5 and 11 other personality inventories, such as the AB5C or JPI. The data of this subset consisted of 966 items and 1,142 respondents, implying 1,304,164 item pairs. From this, we drew a sample of 110,979 pairs to complement the 89,021 pairs implied by the other datasets, amounting to a total of 200,000 training examples used to fine-tune our model.

Fourth, we assessed the generalizability of our approach to detect candidate jingle–jangle fallacies using an additional dataset of items and scale labels from ref. 15. We report those analyses in the Supplementary Information.

## Embeddings

We compared several different language models in our work. We selected these models because they cover a wide range of approaches that can be used to generate linguistic embeddings, spanning simpler models that consider only single words to those that consider several words in the form of sentences and thus promise to be more sensitive to information concerning word order, semantic context and other syntactic information.

In the main article, we use and report results based on embeddings from a fine-tuned model because of its superior performance relative to other approaches. Our approach was inspired by the recent work suggesting that fine-tuning can produce better results than off-the-shelf models for the prediction of psychological phenomena (for example, ref. 37), including the psychometric properties of items and scales[26]. Specifically, we fine-tuned MPNet[20], a lightweight sentence transformer model based on the BERT architecture. We then compared this to other transformer models, including mixedbread (mxbai-embed-large-v1)[38], OpenAI's latest model (text-embedding-3-large)[39] and models relying on other architectures[15,24]. The results for these models and associated comparisons are reported in the Supplementary Information.

The fine-tuned model is publicly available at https://huggingface. co/dwulff/mpnet-personality and a step-by-step tutorial on using Hugging Face models, which includes the use of embeddings to estimate the relation between psychological items, is provided in ref. 18.

**Item embeddings.** We retrieved item embeddings by directly encoding the item texts in the model call.

**Scale embeddings.** We generated scale-level embeddings by summing the item embeddings according to

$$\mathbf{w}_{\text{scale},j} = \sum_{j} \mathbf{w}_{ij} \qquad (1)$$

where $\mathbf{w}_{ij}$ is the embedding vector of the $i$th item in the $j$th scale.

**Label embeddings.** We considered several types of label embeddings that we reasoned could have different strengths and weaknesses. All embeddings were generated for the individual single-construct labels rather than the multi-construct compounds to be able to use them to produce a simplified scale–label mapping (see 'Minimizing jingle–jangle fallacies to increase conceptual clarity and parsimony in psychology').

First, for the base embedding, we directly encoded the IPIP construct labels (for example, 'extraversion' or 'sociability'). However, many construct labels can be used in everyday language without explicit reference to the personality construct (for example, 'warmth'); consequently, we reasoned that additional context could be helpful to appropriately capture the meaning of the personality construct and therefore tested different ways to provide context, leading to several other label embeddings. Second, we produced a contextualized version of the construct labels by placing the construct label in the sentence: 'The personality construct [LABEL].' We then used this sentence to generate embeddings from the language models. Third, we scraped definitions of the available personality constructs from the APA dictionary (https://dictionary.apa.org/). Construct definitions were available for 135 of a total of 277 distinct constructs. Fourth, we manually curated the APA definitions by removing parts of the definitions that referred to meanings other than the personality construct (for example, reference to special meanings of the construct 'competence' in linguistics and law) and filling in text for constructs that were defined exclusively by providing reference to other constructs using the definitions of the other constructs (for example, replacing 'see aggression' with the definition of 'aggression' for the construct 'hostile-aggression'). Fifth, we generated definitions from GPT-4 via the application programming interface (API) using a range of different prompt structures, varying the length (30, 50 or 100 words), assistant instruction (for example, starting or not starting the prompt with: 'You are an expert in psychology and will be asked to produce a definition for a construct that other experts in the field will recognize as accurate and representative.') and core prompt ('Write a [LENGTH]-word expert definition of the personality construct [LABEL].'). We then used the same strategy as for item embeddings described above to generate label embeddings for the different label variations.

## Similarity

The similarity between items, scales and scale labels was calculated as the cosine similarity between embedding vectors and scaled to match the average correlations obtained from empirical responses. Specifically, the cosine similarity (cos) between a vector **a** and **b** was calculated as

$$\cos = \left( \frac{\mathbf{a} \cdot \mathbf{b}}{\|\mathbf{a}\|\|\mathbf{b}\|} \right)$$

## Cronbach's $\alpha$

We obtained the within-scale similarity $S_{jk} = \frac{1}{N_{jk}} \sum S_{ijk}$ with $S_{ijk}$ being the similarity of a pair of items $i$ in scale $j$ of inventory $k$ and $N_{jk}$ being the number of items in the same scale. We then used the Spearman–Brown prediction formula to derive estimates of the scale's Cronbach's $\alpha$ internal consistency measure for scale $j$ in inventory $k$ as follows

$$\hat{\alpha}_{jk} = \frac{N_{jk} S_{jk}}{1 + (N_{jk} - 1) S_{jk}}$$

## Structural fidelity

We calculated structural fidelity for each scale $j$ in inventory $k$ by comparing the similarity of items from a given scale to the similarity of items from different scales from the same inventory as follows

$$\text{fidelity}_{jk} = \frac{S_{jk} - \frac{1}{N_{jk}} \sum_j S_{jk}}{\sqrt{\frac{1}{N_k - 1} \sum_j \left( S_{jk} - \frac{1}{N_k} \sum_j S_{jk} \right)^2}}$$

## Scale–label alignment

We calculated the alignment between scales and labels by comparing the average similarity between a scale's items and the assigned label to the average similarities with other possible labels:

$$\text{alignment}_{jk} = \frac{S_{jl} - \frac{1}{N_{jl}} \sum_j S_{jl}}{\sqrt{\frac{1}{N_l - 1} \sum_j \left( S_{jl} - \frac{1}{N_l} \sum_j S_{jl} \right)^2}}$$

with $l$ indexing labels.

## Reporting summary

Further information on research design is available in the Nature Portfolio Reporting Summary linked to this article.

## Data availability

The data of study this study are available at https://osf.io/nmv29/.

## Code availability

The materials of study this study are available at https://osf.io/nmv29/. Analyses were performed using the R programming language (v.4.3.1) and Python (v.3.9.16).

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

## Acknowledgements

This work was supported by grants from the Swiss National Science Foundation to D.U.W. (100015_197315) and R.M. (100015_204700).

The funders had no role in study design, data collection and analysis, decision to publish or preparation of the manuscript. We thank A. Bagaïni and Z. Hussain for helpful comments and L. Wiles for editing the manuscript.

## Author contributions

D.U.W. and R.M. conceptualized the study and wrote the original draft. D.U.W. was responsible for formal analysis.

## Funding

## Competing interests

The authors declare no competing interests.

## Additional information

**Correspondence and requests for materials** should be addressed to Dirk U. Wulff.

# Reporting Summary

## Statistics

For all statistical analyses, confirm that the following items are present in the figure legend, table legend, main text, or Methods section.

| n/a | Confirmed | |
|---|---|---|
| ☐ | ☒ | The exact sample size ($n$) for each experimental group/condition, given as a discrete number and unit of measurement |
| ☐ | ☒ | A statement on whether measurements were taken from distinct samples or whether the same sample was measured repeatedly |
| ☒ | ☐ | The statistical test(s) used AND whether they are one- or two-sided <br> *Only common tests should be described solely by name; describe more complex techniques in the Methods section.* |
| ☐ | ☒ | A description of all covariates tested |
| ☐ | ☒ | A description of any assumptions or corrections, such as tests of normality and adjustment for multiple comparisons |
| ☐ | ☒ | A full description of the statistical parameters including central tendency (e.g. means) or other basic estimates (e.g. regression coefficient) AND variation (e.g. standard deviation) or associated estimates of uncertainty (e.g. confidence intervals) |
| ☒ | ☐ | For null hypothesis testing, the test statistic (e.g. $F$, $t$, $r$) with confidence intervals, effect sizes, degrees of freedom and $P$ value noted <br> *Give P values as exact values whenever suitable.* |
| ☒ | ☐ | For Bayesian analysis, information on the choice of priors and Markov chain Monte Carlo settings |
| ☐ | ☒ | For hierarchical and complex designs, identification of the appropriate level for tests and full reporting of outcomes |
| ☐ | ☒ | Estimates of effect sizes (e.g. Cohen's $d$, Pearson's $r$), indicating how they were calculated |

*Our web collection on statistics for biologists contains articles on many of the points above.*

## Software and code

Policy information about availability of computer code

| Data collection | No software was used for data collection. |
|---|---|
| Data analysis | The codes used to analyze the data are publicly available in an online repository (https://osf.io/nmv29/). Analyses were performed using the R programming language (version 4.3.1) and Python (version 3.9.16). |

For manuscripts utilizing custom algorithms or software that are central to the research but not yet described in published literature, software must be made available to editors and reviewers. We strongly encourage code deposition in a community repository (e.g. GitHub). See the Nature Portfolio guidelines for submitting code & software for further information.

## Data

Policy information about availability of data

All manuscripts must include a data availability statement. This statement should provide the following information, where applicable:
- Accession codes, unique identifiers, or web links for publicly available datasets
- A description of any restrictions on data availability
- For clinical datasets or third party data, please ensure that the statement adheres to our policy

Data used in our work (personality measures and data from IPIP and Rosenbusch et al., 2020) is made publicly available by the original data providers and the web links are directly provided in the manuscript. We make the curated data for IPIP labels and APA definitions available in our public repository (https://osf.io/nmv29/).

# Research involving human participants, their data, or biological material

Policy information about studies with human participants or human data. See also policy information about sex, gender (identity/presentation), and sexual orientation and race, ethnicity and racism.

| | |
|---|---|
| Reporting on sex and gender | We do not make use of sex or gender information in our analysis. |
| Reporting on race, ethnicity, or other socially relevant groupings | We did not include race or ethnicity in our analyses. However, in our validation of item and scale embeddings we use personality data for and English-speaking US-based group because of our focus on English-based embeddings of personality items and construct definitions. |
| Population characteristics | We use personality data from a US sample and several citizen-science data sets from the Open Psychometrics Project to validate item and scale embeddings. |
| Recruitment | We did not recruit participants for this study. We used data from existing data sets. |
| Ethics oversight | *Identify the organization(s) that approved the study protocol.* |

Note that full information on the approval of the study protocol must also be provided in the manuscript.

# Field-specific reporting

Please select the one below that is the best fit for your research. If you are not sure, read the appropriate sections before making your selection.

☐ Life sciences   ☒ Behavioural & social sciences   ☐ Ecological, evolutionary & environmental sciences

For a reference copy of the document with all sections, see nature.com/documents/nr-reporting-summary-flat.pdf

# Behavioural & social sciences study design

All studies must disclose on these points even when the disclosure is negative.

| | |
|---|---|
| Study description | The study is quantiative in nature. It involved using embeddings of linguistic data from personality measures that are publicly available and comparison to aggregate results based on human data. |
| Research sample | For this study we used existing datasets and did not recruit human research participants. The descriptions and sources are provided in the methods section. Data used in our work (personality measures and data from IPIP and Rosenbusch et al., 2020). |
| Sampling strategy | We included all available data concerning items (International Personality Item Pool and Data of Rosenbusch et al., 2020) and available construct definitions (American Psychological Association Dictionary) in our analyses. In addition, we used personality data from existing data repositories (Johnson, 2014, 2020; Open Psychometrics Project). |
| Data collection | We did not directly collect data from human research participants but analyzed data from existing data sets. |
| Timing | We included data that was available as of July 1, 2024. |
| Data exclusions | For our validation analyses (e.g., comparison of psychometric characteristics based on LLMs to those of human data) we selected personality data from English-speaking population only (Johnson, 2014; Open Psychometrics Project) given our focus on English personality items (IPIP) and definitions (APA). |
| Non-participation | In this study we did not directly collect data from human research participants and therefore our protocol does not describe or address non-participation. |
| Randomization | Participants were not allocated to experimental groups. |

# Reporting for specific materials, systems and methods

We require information from authors about some types of materials, experimental systems and methods used in many studies. Here, indicate whether each material, system or method listed is relevant to your study. If you are not sure if a list item applies to your research, read the appropriate section before selecting a response.

## Materials & experimental systems

| n/a | Involved in the study |
|-----|----------------------|
| ☒ ☐ | Antibodies |
| ☒ ☐ | Eukaryotic cell lines |
| ☒ ☐ | Palaeontology and archaeology |
| ☒ ☐ | Animals and other organisms |
| ☒ ☐ | Clinical data |
| ☒ ☐ | Dual use research of concern |
| ☒ ☐ | Plants |

## Methods

| n/a | Involved in the study |
|-----|----------------------|
| ☒ ☐ | ChIP-seq |
| ☒ ☐ | Flow cytometry |
| ☒ ☐ | MRI-based neuroimaging |

## Plants

| | |
|---|---|
| Seed stocks | none |
| Novel plant genotypes | none |
| Authentication | none |

