## [Peer Review File · Nature Human Behaviour]

Semantic embeddings reveal and address taxonomic incommensurability in psychological measurement

Corresponding Author: Dr Dirk Wulff

Version 0:

Decision Letter:

23th November 2023

Dear Dr Wulff,

Thank you once again for your manuscript, entitled "Automated jingle-jangle detection: Using embeddings to tackle taxonomic incommensurability", and for your patience during the peer review process.

Your Article has now been evaluated by 3 referees. You will see from their comments copied below that, although they find your work of potential interest, they have raised quite substantial concerns. In light of these comments, we cannot accept the manuscript for publication, but would be interested in considering a revised version if you are willing and able to fully address reviewer and editorial concerns.

We hope you will find the referees' comments useful as you decide how to proceed. If you wish to submit a substantially revised manuscript, please bear in mind that we will be reluctant to approach the referees again in the absence of major revisions. We are committed to providing a fair and constructive peer-review process. Do not hesitate to contact us if there are specific requests from the reviewers that you believe are technically impossible or unlikely to yield a meaningful outcome.

To guide the scope of the revisions, the editors discuss the referee reports in detail within the team, including with the chief editor, with a view to (1) identifying key priorities that should be addressed in revision and (2) overruling referee requests that are deemed beyond the scope of the current study. We hope that you will find the prioritized set of referee points to be useful when revising your study. Please do not hesitate to get in touch if you would like to discuss these issues further.

In particular, we ask that you:

- 1) prove the generalizability of your approach and findings by extending the analyses to a larger corpus (as suggested by Reviewer #3).
- 2) present your work in the context of prior research, clarifying its contribution and broader implications (as requested by Reviewers #2 and #1).

If you wish to submit a suitably revised manuscript, we would hope to receive it within 4 months. I would be grateful if you could contact us as soon as possible if you foresee difficulties with meeting this target resubmission date.

- Include a "Response to the editors and reviewers" document detailing, point-by-point, how you addressed each editor and referee comment. If no action was taken to address a point, you must provide a compelling argument. When formatting this document, please respond to each reviewer comment individually, including the full text of the reviewer comment verbatim followed by your response to the individual point. This response will be used by the editors to evaluate your revision and sent back to the reviewers along with the revised manuscript.
- Highlight all changes made to your manuscript or provide us with a version that tracks changes.

Link Redacted

Thank you for the opportunity to review your work. Please do not hesitate to contact me if you have any questions or would like to discuss the required revisions further.

[redacted]

Reviewer expertise:

Reviewer #1: ML, computational linguistics, social science

Reviewer #2: Psychological constructs, personality, NLP

Reviewer #3: Psychological constructs, personality, ML

REVIEWER COMMENTS:

Reviewer #1:

Remarks to the Author:

I think the fundamental idea of this paper, using embeddings of survey items to address the jingle/jangle fallacy is excellent. And what I see as the main result, figure 5, is very nice.

That said, I think the paper could be stronger both in terms of clarity (See some easily addressed questions below) and in terms of explanation.

I'm not sure figure 2 adds much; it just shows that you do compute lots of correlations. (but it is clear, so I leave it up to the authors whether to keep it)

I found figure 3 hard to understand

Figure 3C - what is the "signed correlation"- if it were in color I could see the signs, but in black and white I don't know what the values range over.

Figure 3D what are the units of "Scale-label alignment"? The text refers to cosign similarity but what does e.g. a "3" mean? Isn't this a similarity that can't exceed 1?

Figure 3B What is "structural fidelity" and "empirical fidelity"? I think the relevant text is as follows, but I still can't picture exactly what the z-score in question is. (and z-scores of 5 seem high?)
An equation would help

"We evaluated the structural fidelity of scales by evaluating the similarity of items with the other items in the same scale in comparison to the similarity of the scale's items to the items of the other scales. Specifically, we computed a z-score for each scale that reflects the similarity of items within the same scale relative to the similarity of items of different scales within an inventory."

I liked the observation that "A large number of the 60 jingle fallacies (58%) concern the ORVIS and ORAIS inventories,..." but I'm not entirely clear if this is showing a problem with the scales or with the embeddings. I think the authors are implying it is the scales, but a couple sentences might clarify that.

I wonder if it would be possible to give more concrete specific examples, for example

I would love to see an embedding based version of the following paper. (but I understand this specific desire is out of scope for this paper)

Weidman, Aaron C., Conor M. Steckler, and Jessica L. Tracy. "The jingle and jangle of emotion assessment: Imprecise measurement, casual scale usage, and conceptual fuzziness in emotion research." *Emotion* 17.2 (2017): 267.

but a couple more little case studies (one paragraph each) of insights would be helpful.

Or some bigger picture discussion about how this relates to the recent papers that show that the 5-factor

model does an amazingly good job capturing a huge number of scales.

Finally, I'm curious what you found about reverse coding, which we see as being handled significantly differently by embeddings.

Reviewer #2:

Remarks to the Author:

Review of "Automated jingle-jangle detection: Using embeddings to tackle taxonomic incommensurability."

I am grateful for the opportunity to comment on this paper. It can become an essential paper in the high-potential algorithmic/psychometric integration literature.

This paper proposes that so-called "taxonomic incommensurability" is holding theories back due to the jingle-jangle fallacies (constructs with the same or different names representing different and the same underlying constructs, respectively). It suggests that neural network embeddings may address the jingle-jangle fallacies and the prediction of empirical relationships between constructs.

While taking on an important subject, this paper has missed a whole raft of existing literature, making it less novel than the authors suggest. Had this literature been incorporated, the paper may still have made a good contribution, as no other paper I know has reached this accuracy level.

The missed literature includes the first paper to address jingle-jangle fallacies through embeddings or semantic spaces. The paper was published in the top journal of the information systems discipline:

Larsen, Kai R., and Chih How Bong. "A tool for addressing construct identity in literature reviews and meta-analyses." *MIS Quarterly* 40.3 (2016): 529-552.

In terms of predicting correlations using embeddings or semantic spaces, this area was pioneered by Arnulf et al. almost a decade ago, and there must be another ten papers or so for the "semantic theory of survey response" by now:

Arnulf, Jan Ketil, et al. "Predicting survey responses: How and why semantics shape survey statistics on organizational behaviour." *PloS one* 9.9 (2014): e106361.

A backward chaining search in Google Scholar should reveal this whole literature.

Until the authors incorporate these two research areas, there is no meaningful way to evaluate their paper's contributions. However, I suggest that the next step to move this project forward to a meaningful contribution in this era of AI would be to move from cosines to supervised machine learning. Further, the authors are operating with personality data, which is an area in which they could define a unique contribution because Arnulf et al. (2014), while they showed that they could predict correlational patterns of a survey, were not able to replicate those findings for personality scales.

Specific feedback on evaluations:

1. Correlations between items from the same scale (Cronbach's Alpha): I'm not sure I've seen anyone else do this before, but it is a pretty obvious analysis that, due to its focus on items that measure (or belong to) the same underlying constructs, one would expect a high r . The relatively low correlations obtained (.28 - .53) are probably due to the focus on the personality domain, where scales were initially built differently than most other types of psychological theory.
2. Recovering the scales' structural fidelity, the authors consider a scale fully recovered when the z-score for each scale is above 2. This will require better justification and support from the literature.
3. The authors employ another cutoff point without justifying it for automated jingle-jangle detection. The authors state as their cutoff point that a cosine difference of more than four times the average absolute difference between label and scale similarity and nearly three times the average error in predicting empirical scale correlations represents a meaningful cutoff. Again, without justification, evaluation, or citations. It is worth mentioning that cosine scores are non-linear, which will affect the cutoff. I would recommend converting the cosines to a linear scale before introducing such a cutoff.
4. Also, for automated jingle-jangle detection, given the cutoff point the authors specify in my point #3, above, they then evaluate the success of the algorithm. This means that they only evaluate true positives vs. false positives, but miss the opportunity to evaluate true negatives vs. false negatives. They identify 60 jingle and 53 jangle fallacies, which is fewer jangle fallacies than I expected. Other research (such as Larsen and Bong) has shown that there are many times more jangle fallacies than jingle fallacies. It is possible this comes from the authors' more careful measurement of construct name similarity than that employed by Larsen and Bong, who used direct name match for this part of their evaluation. On the other hand, Larsen and Bong seem to have had access to a fully categorized set of constructs, which allowed them to address both type 1 and type 2 errors in their algorithms.

The solutions section of the paper is the one that most excites me. While cluster algorithms are notoriously unreliable and cosine similarities leave most of the variance unaccounted for (relative to supervised machine learning), this section is moving in a highly productive direction. I wish the authors had been more thorough in their literature review, as without it, it is hard to be sure of the actual contributions of the manuscript.

Minor typo: "and generated definitions form GPT-4."

Reviewer #3:

Remarks to the Author:

The paper provides very insightful demonstrations of using LLM embeddings to identify (and even address) jingle-jangle fallacies in psychological measurements. I have only a few comments.

- 1) I think readers might need more guidance as to what they should do now. It appears that the paper—despite being very creative in the presented analyses—limits itself to be a demonstration for now. I wonder whether it can have more practical impact, maybe through policy suggestions for journals (e.g., maintenance of an openly searchable semantic space for scales), or tutorial materials, or an improved version of the online app in the Rosenbusch et al paper. I guess the latter would need users to have an openai api key but the costs for encoding a single scale are minimal.
- 2) The ipip data is great because it is easily accessible but they only constitute a very small part of the scale jungle in psychology. If the authors applied their approach to a larger corpus (e.g., the one of Rosenbusch et al.) that would not only lead to new insights but also allow for a sort of cross-validation of the decisions made for the ipip data. Plus it would offer a semantic space that scale developers and reviewers could query for existing scales.
- 3) I would like to hear the authors' thoughts on the case when both scale content and label are highly similar. This is neither a jingle nor a jangle fallacy, but it is still bad because of redundancy. So why not suggest cutting all scales with similar contents, regardless of their labels?
- 4) I think the term taxonomic incommensurability might not be the most accessible. I would suggest omitting or changing it for easy understandability and improved twitter marketing 😊
- 5) There were moments towards the beginning of the manuscript where I thought 'labels' referred to the words at the poles of item rating scales (so labels would be things like "strongly agree" or "not at all interested")
- 6) I looked for an in-text link for supplementary materials and possibly pre-registration materials but I couldn't find any.

Hannes Rosenbusch

Version 1:

Decision Letter:

3rd June 2024

Dear Dr Wulff,

Thank you once again for your revised manuscript, entitled "Using embeddings to automate jingle--jangle detection and tackle taxonomic incommensurability," and for your patience during the re-review process.

Your manuscript has now been evaluated by the same reviewers who evaluated your original manuscript. All reviewer feedback is included at the end of this letter. Although the reviewers found your manuscript to have improved during revision, they also raise some important outstanding concerns. We remain interested in the possibility of publishing your study in Nature Human Behaviour, but would like to consider your response to these outstanding concerns in the form of a revised manuscript before we make a decision on publication.

In particular, we ask that you [1] provide the additional information asked by Reviewer #1, and [2] improve the discussion of the existing literature applying language models to the jingle-jangle fallacy, following the guidance provided by Reviewer #2. Please note that this is a final opportunity to address the reviewers' comments.

In sum, we invite you to revise your manuscript taking into account all reviewer and editor comments. We are committed to providing a fair and constructive peer-review process. Do not hesitate to contact us if there are specific requests from the reviewers that you believe are technically impossible or unlikely to yield a meaningful outcome.

We hope to receive your revised manuscript within 4-8 weeks. I would be grateful if you could contact us as soon as possible if you foresee difficulties with meeting this target resubmission date.

- Include a "Response to the editors and reviewers" document detailing, point-by-point, how you addressed each editor and referee comment. If no action was taken to address a point, you must provide a compelling argument. This response will be used by the editors and reviewers to evaluate your revision.
- Highlight all changes made to your manuscript or provide us with a version that tracks changes.

Link Redacted

We look forward to seeing the revised manuscript and thank you for the opportunity to review your work. Please do not hesitate to contact me if you have any questions or would like to discuss these revisions further.

[redacted]

Reviewer expertise:

Reviewer #1: ML, computational linguistics, social science

Reviewer #2: Psychological constructs, personality, NLP

Reviewer #3: Psychological constructs, personality, ML

REVIEWER COMMENTS:

Reviewer #1:

Remarks to the Author:

The authors have done a good job of adding in more prior literature and comparisons.

I think they have most of the content that they need, but

I still find the article frustratingly hard to read.

Terms are often used without definition until much later in the article.

It is fine if some of the figures (especially in the supplement) are not important, but the key results are often hard to find.

For example, I don't understand the authors' answer to my previous question "I don't know what the values range over. Figure 3D what are the units of "Scale-label alignment"? The text refers to cosine similarity but what does e.g. a "3" mean? Isn't this a similarity that can't exceed 1?" This is now panel F, and the legend says "Panel F illustrates our construct label evaluation. The points show the average alignment score as defined in the text ..." I believe you, but after a few minutes of searching in the text for "alignment" I gave up. Please put it where I can easily find it. For example when you first use the term.

Similarly, the article talks a lot about embeddings, but the introduction doesn't say what sort of embeddings they are. (Basically, a variation of RoBERTa, which one can guess from the 768 in the figure caption, or by going to the supplemental material, or by reading patiently until page 11, when you mention "Specifically, we fine-tuned MPNet [22]"--but don't say what MPNet is.) Please put a paragraph in the main text saying a little about what embedding you used and briefly explain *why* you used that embedding? Since you are computing similarities, I would have assumed that a model like sbert would have been better. Or that if you are fine-tuning, then maybe any RoBERTa-class model is pretty much equivalent?

The supplement is equally hard to read. Can you state clearly (or maybe I just missed it) what is being done for fine-tuning? You are adjusting what (all the parameters in MPNet?) to reduce what loss function?

You concluded that your fine-tuned model is better than the other ones. That makes sense, but can you point me to (and cite in the main paper) the table that shows the numbers (e.g. correlations) for each of the embeddings that you used to reach that conclusion.

I'm also confused by some of the notation. E.g.,
"At the level of items, we observed extremely low mean absolute errors between $r = .026$ (FFM) and $r = .029$ (16PF) in-sample and higher but still low errors between $r = .07$ (BIG5) and $r = .065$ (HEXCO) out-of-sample. These imply correlations of, on average, $r = .96$ (in-sample) and $r = .64$ (out-of-sample)."
- I'm a bit confused by a "mean absolute error" being represented by the letter r , especially since the same paragraph uses r to be a correlation (which makes more sense to me). I think the MAE is something entirely different?

Finally, in the supplement, equation 2, what is the norm of n squared? You are doing a weighted sum in the numerator, weighting, but n_i , so would expect you would divide by $\|n\|$, and not square it.

Reviewer #2:

Remarks to the Author:

Review of "Using embeddings to automate jingle-jangle detection and tackle taxonomic incommensurability"

As a staunch advocate of the necessity for research on taxonomic incommensurability, I firmly believe that a paper that effectively represents this research area not only deserves a place in *Nature: Human Behavior* but also contributes significantly to the advancement of our field.

A paper must offer a unique contribution to be published in a top journal. In the initial version of the paper, the authors proposed an approach to address taxonomic incommensurability, suggesting it was the first to tackle the jingle-jangle fallacy using large language models.

As should be apparent from the last round of review, the authors did not invent large language models and were not the first to use language models to address the jingle-jangle fallacy. The question is whether using a "large" language model rather than a "medium" language model represents novelty. Those of us who have worked in the area of language models for a few decades would probably agree that the computer scientists who developed innovations such as the attention mechanism (Vaswani et al. 2007) provided true novelty, but employing a large language model to an existing research area does not in itself represent novelty over existing work also employing language models to the same problem.

In short, we would likely not accept a paper that employed a new statistical approach (not developed by the authors themselves) to address self-efficacy without deeply engaging with the existing literature on self-efficacy (such as Bandura's work). Given that the authors are not the first to address the jingle-jangle fallacy using language models, they must much more extensively engage with the relevant literature to explain the true novelty of their work over the existing work. Anything else would come across as academically dishonest and not in line with publication in a top journal. If the authors are unwilling to engage with the existing literature, this paper should not be published in *NHB*. I'm hoping the authors will be allowed to attempt this, though I've been discouraged after seeing two versions of their paper. As I have been requested by the editor of *NHB* to provide more information on the existing literature, I have tried to do that in the following.

The paper by Larsen and Bong (2016), published in the top journal of the Information Systems discipline, represented the first effort to address what they termed construct identity fallacies (a combination of the jingle fallacy and the jangle fallacy in a 2x2 framework) in behavioral and social sciences using natural language processing (NLP) techniques. Their work, particularly the Construct Identity Detector (CID) algorithm, which incorporates latent semantic analysis (LSA) and improves it, sets a foundational precedent for detecting construct overlaps and differentiations. LSA was one of two vector space models examined and extended. As contemporary large language models (LLMs) like GPT-4 are scrutinized for their advancements, examining how these models are technically analogous to, yet an evolution of, the LSA-based methods described by Larsen and Bong is imperative. For example, given that Larsen and Bong (2016) calculated the ROC AUC's for their models, I think that is all you need to do for comparison and to prove that your models are better. It would be shocking if they weren't, but we would learn from knowing how much better without adding much extra effort.

Technical Parallels

1. Foundational Principles: LLMs and LSA are grounded in the vector space model. LSA reduces the dimensionality of text data to capture semantic meaning through singular value decomposition (SVD), a form of matrix factorization. Similarly, through transformer architectures, LLMs represent text in high-dimensional space using embeddings that encapsulate contextual semantics. In fact, it is often possible to treat LLM embeddings as LSA vectors and vice versa.

2. Semantic Understanding: LSA captures word meanings based on their co-occurrence patterns in large corpora. LLMs extend this by training on massive datasets and using attention mechanisms to understand context over long text spans, improving semantic capture and context awareness.

3. Application in Construct Identity: Larsen and Bong's CID1 combines LSA and other NLP algorithms to detect construct identities. LLMs can perform similar tasks more efficiently due to their training on diverse datasets, enabling them to recognize nuanced semantic relationships without extensive domain-specific tuning. This is worth mentioning as part of the justification for why your paper is worth publishing.

Contributions of Larsen and Bong (2016)

1. Construct Identity Detector (CID): Larsen and Bong introduced CID, a tool that combines LSA with other algorithms to detect construct identity fallacies. This was a significant step towards addressing the jingle and jangle fallacies by providing an automated method to evaluate construct identities against human expert decisions.
2. Framework for Evaluating NLP Designs: They established a rigorous framework for comparing different NLP designs, demonstrating that hybrid models could outperform single-method approaches in construct detection tasks.
3. Implications for Behavioral Sciences: Their work highlighted the importance of addressing construct identity for the advancement of behavioral sciences, emphasizing the necessity for tools that can assist in literature reviews and meta-analyses by identifying construct overlaps. This is an area where a paper like yours would be expected to build further and carefully explain how the new model outperforms LSA and what this difference means for psychometric practice. It would benefit the authors to read the behavioral implications that came out of the Larsen and Bong paper and discuss whether their own work has additional implications.

While LLMs represent a significant technical advancement over traditional methods like LSA, the foundational principles remain similar. The authors do not claim to have built the original LLMs or contributed to their algorithmic development by inventing the attention mechanism, and they are extending an existing finding, the work of Larsen and Bong (2016) provides a crucial stepping stone for describing the authors' true contributions. Likely, their contribution lies in showing that they can now address construct identity fallacies more effectively. By engaging LLM capabilities, the authors have likely developed more accurate and efficient tools for construct identity detection, thereby contributing to enhancing the rigor and reliability of literature reviews and meta-analyses in social and behavioral sciences.

Reviewer #3:

Remarks to the Author:

Thank you for a very interesting paper. All my prior remarks have been addressed and I enjoyed reading the revised manuscript.

Best wishes,

Hannes Rosenbusch

Version 2:

Decision Letter:

Our ref: NATHUMBEHAV-23103404B

5th September 2024

Dear Dr Wulff,

Thank you for submitting your revised manuscript "Using embeddings to automate jingle-jangle detection and tackle taxonomic incommensurability" (NATHUMBEHAV-23103404B). It has now been seen by two of the original referees and their comments are below. As you can see, the reviewers find that the paper has improved in revision. We will therefore be happy in principle to publish it in Nature Human Behaviour, pending revisions to satisfy the referees' final requests and to comply with our editorial and formatting guidelines.

We are now performing detailed checks on your paper and will send you a checklist detailing our editorial and formatting requirements within two weeks. Please do not upload the final materials and make any revisions until you receive this additional information from us.

[redacted]

Reviewer #1 (Remarks to the Author):

the authors have adequately responded to all of my requests for clarification.

Reviewer #2 (Remarks to the Author):

The authors have done a good job in responding to the comments of the last round.

From my point of view, the authors have done enough that I have no further negative comments. The only concerns I might rise at this point is that some of the descriptive language is still convoluted and probably not possible to follow for lay-people with little experience in language models or jingle jangle fallacies. A clear example is figure 1 and its description.

Reviewer #2 (Remarks on code availability):

I was able to run the code and test out their conclusions.

Dear Editor,

Thank you for the input provided and the opportunity to revise our manuscript (previously) entitled “Automated jingle-jangle detection: Using embeddings to tackle taxonomic incommensurability” for publication in Nature Human Behavior.

As we detail below, we have addressed all points raised by you and the reviewers. Most centrally, we 1) demonstrate the generalizability of our approach by extending our analysis to a larger corpus of constructs and items and 2) rewrote several portions of our work to better place it in the context of prior research.

In addition, we would like to note that we have further strengthened our contribution by introducing a novel fine-tuned model that outperforms an updated set of state-of-the-art pretrained models for the prediction of key psychometric properties of personality measures and, therefore, provides an even stronger basis for our automated jingle-jangle detection. We now also make this model publicly available to help future efforts addressing conceptual clarity in psychology.

We very much appreciate your and the reviewers’ help with improving our contribution and hope you agree that our improved manuscript is now even better placed to help advance conceptual clarity in the psychological and behavioral sciences.

Kind regards,

Dirk Wulff and Rui Mata

Editor:

Your Article has now been evaluated by 3 referees. You will see from their comments copied below that, although they find your work of potential interest, they have raised quite substantial concerns. In light of these comments, we cannot accept the manuscript for publication, but would be interested in considering a revised version if you are willing and able to fully address reviewer and editorial concerns.

We hope you will find the referees' comments useful as you decide how to proceed. If you wish to submit a substantially revised manuscript, please bear in mind that we will be reluctant to approach the referees again in the absence of major revisions. We are committed to providing a fair and constructive peer-review process. Do not hesitate to contact us if there are specific requests from the reviewers that you believe are technically impossible or unlikely to yield a meaningful outcome.

We appreciate the overall positive assessment of our work, the constructive input, and the opportunity to revise our manuscript.

To guide the scope of the revisions, the editors discuss the referee reports in detail within the team, including with the chief editor, with a view to (1) identifying key priorities that should be addressed in revision and (2) overruling referee requests that are deemed beyond the scope of the current study. We hope that you will find the prioritized set of referee points to be useful when revising your study. Please do not hesitate to get in touch if you would like to discuss these issues further.

In particular, we ask that you:

1) prove the generalizability of your approach and findings by extending the analyses to a larger corpus (as suggested by Reviewer #3).

We have now expanded our approach by including an additional analysis using data from Rosenbusch et al. (2020). This data set includes a considerably larger set of items and scales and covers a broader range of topics, including personality and attitudinal domains, but also others, such as religious and political practices, eating behavior, and clinical domains. Crucially, we replicate the central findings from our analysis using IPIP data. Specifically, we find evidence of both jingle and jangle fallacy candidates and a key trade-off between the two. Further, we show that the relabeling approach can produce a more parsimonious version of a construct taxonomy and associated measures. We should note that the Rosenbusch et al. data is not without limitations, including a limited curation of label and scale names relative to the IPIP data set. As a result, and also for ease of exposition, despite the larger size of the Rosenbusch et al. data set, we opted to keep the IPIP analysis in the main text and document the new analysis of the Rosenbusch et al. data in the supplementary materials.

2) present your work in the context of prior research, clarifying its contribution and broader implications (as requested by Reviewers #2 and #1).

We now have rewritten portions of the introduction and discussion to clarify our contribution relative to prior research. In particular, we now make clearer the contributions of our approach using novel large language models relative to previous attempts that are similar in spirit but have used other techniques (latent semantic analysis models) and did not consider crucial aspects of conceptual clarity, such as the semantic overlap of construct labels. Further, we point out further novel contributions of our work, including the validation of scale labels, the introduction of new evaluation metrics (structural fidelity), and, perhaps more importantly, the proposal of using a relabeling procedure to achieve more parsimonious taxonomies of constructs and associated measures.

Finally, concerning broader implications, we discuss in more detail how our approach can be utilized in conjunction with several approaches, including expert consensus and other scientific practices (e.g., journal policies), to ensure that conceptual clarity improves both in psychology and other disciplines.

Reviewer #1:

I think the fundamental idea of this paper, using embeddings of survey items to address the jingle/jangle fallacy is excellent. And what I see as the main result, figure 5, is very nice. That said, I think the paper could be stronger both in terms of clarity (See some easily addressed questions below) and in terms of explanation.

Thank you for the overall positive assessment. We appreciate the concrete input provided that has helped us rewrite several sections to improve the clarity of our exposition.

I'm not sure figure 2 adds much; it just shows that you do compute lots of correlations. (but it is clear, so I leave it up to the authors whether to keep it)

We understand the reviewer's comment but we believe the figure helps illustrate our approach for readers with less experience with embedding models and makes clear from the outset that we work with several types of embeddings (item, scale, label). We have now relabeled the figure title accordingly.

I found figure 3 hard to understand. Figure 3C - what is the "signed correlation"- if it were in color I could see the signs, but in black and white I don't know what the values range over. Figure 3D what are the units of "Scale-label alignment"? The text refers to cosign similarity but what does e.g. a "3" mean? Isn't this a similarity that can't exceed 1?

To deal with the specific points raised, we 1) modified the figure legend to provide an explicit definition of structural fidelity ("defined as the z-score contrasting the similarity of items within a given scale to items of others scales within the same inventory") and 2) eliminated the distinction between signed vs unsigned correlations from the figure because it was a minor point that we now only address in writing.

Figure 3B What is "structural fidelity" and "empirical fidelity"? I think the relevant text is as follows, but I still can't picture exactly what the z-score in question is. (and z-scores of 5 seem high?) An equation would help. "We evaluated the structural fidelity of scales by evaluating the similarity of items with the other items in the same scale in comparison to the similarity of the scale's items to the items of the other scales. Specifically, we computed a z-score for each scale that reflects the similarity of items within the same scale relative to the similarity of items of different scales within an inventory."

We have added an equation in the Methods section detailing how structural fidelity is defined. Empirical fidelity is calculated analogously based on empirical rather than predicted correlations.

I liked the observation that "A large number of the 60 jingle fallacies (58%) concern the ORVIS and ORAIS inventories,..." but I'm not entirely clear if this is showing a problem with the scales or with the embeddings. I think the authors are implying it is the scales, but a couple sentences might clarify that.

The paragraph aimed to describe the candidate jingle and jangle fallacies identified using our method and, therefore, refers to a potential problem with the scales rather than the embeddings. We have rewritten the paragraph to make this explicit and reflect other changes associated with using a more accurate, fine-tuned embedding model.

I wonder if it would be possible to give more concrete specific examples, for example I would love to see an embedding based version of the following paper. (but I understand this specific desire is out of scope for this paper)

*Weidman, Aaron C., Conor M. Steckler, and Jessica L. Tracy. "The jingle and jangle of emotion assessment: Imprecise measurement, casual scale usage, and conceptual fuzziness in emotion research." *Emotion* 17.2 (2017): 267.*

but a couple more little case studies (one paragraph each) of insights would be helpful. Or some bigger picture discussion about how this relates to the recent papers that show that the 5-factor model does an amazingly good job capturing a huge number of scales.

Thank you for pointing out this reference which we think is a great example of an area in which our approach could be fruitfully used. We now cite this paper both in the introduction when first introducing the idea of jingle-jangle fallacies and in the discussion as an area for future application (along with additional examples). In addition, we now provide an analysis of a data set from Rosenbusch et al. (2020) that provides yet a further application of our approach to a wider range of psychological measures.

Finally, I'm curious what you found about reverse coding, which we see as being handled significantly differently by embeddings.

We agree with the reviewer that reverse coding is an interesting issue, but we have decided not to focus on it in our work. Still, we have analyzed the performance of the fine-tuned model for positively (MAE = .032) and negatively (MAE = .03) correlated items and found it to be equivalent.

Reviewer #2:

Review of "Automated jingle-jangle detection: Using embeddings to tackle taxonomic incommensurability." I am grateful for the opportunity to comment on this paper. It can become an essential paper in the high-potential algorithmic/psychometric integration literature. This paper proposes that so-called "taxonomic incommensurability" is holding theories back due to the jingle-jangle fallacies (constructs with the same or different names representing different and the same underlying constructs, respectively). It suggests that neural network embeddings may address the jingle-jangle fallacies and the prediction of empirical relationships between constructs. While taking on an important subject, this paper has missed a whole raft of existing literature, making it less novel than the authors suggest. Had this literature been incorporated, the paper may still have made a good contribution, as no other paper I know has reached this accuracy level.

We would like to thank the reviewer for the input. We agree that it is important to place our work in the larger context of past attempts and have rewritten portions of our introduction and discussion to better acknowledge past work while pointing out the strengths of our contribution relative to previous attempts.

*The missed literature includes the first paper to address jingle-jangle fallacies through embeddings or semantic spaces. The paper was published in the top journal of the information systems discipline: Larsen, Kai R., and Chih How Bong. "A tool for addressing construct identity in literature reviews and meta-analyses." *MIS Quarterly* 40.3 (2016): 529-552. In terms of predicting correlations using embeddings or semantic spaces, this area was pioneered by Arnulf et al. almost a decade ago, and there must be another ten papers or so for the "semantic theory of survey response" by now: Arnulf, Jan Ketil, et al. "Predicting survey responses: How and why semantics shape survey statistics on organizational behaviour." *PloS one* 9.9 (2014): e106361. A backward chaining search in Google Scholar should reveal this whole literature.*

We thank the reviewer for bringing these articles to our attention. We now cite both the Larsen and Bong (2016) and Arnulf et al. (2014) papers in our introduction as further examples of work pointing out the need for eliminating jingle-jangle fallacies and using natural language processing methods for this purpose. Nevertheless, and despite the common goals, we also would like to make clear that this past work has relied on the latent semantic analysis approach akin to the Rosenbausch et al. (2020) paper to predict similarity between measures we had already referenced in our work and used as a comparison model (see Supplementary Materials). As a result, we have opted to cite the suggested papers to provide context but to not provide a full summary of the latent semantic analysis approaches that our work shows are inferior in predicting psychometric characteristics of psychological measures.

Until the authors incorporate these two research areas, there is no meaningful way to evaluate their paper's contributions. However, I suggest that the next step to move this project forward to a meaningful contribution in this era of AI would be to move from cosines to supervised machine learning. Further, the authors are operating with personality data, which is an area in which they could define a unique contribution because Arnulf et al.

(2014), while they showed that they could predict correlational patterns of a survey, were not able to replicate those findings for personality scales.

As discussed in the previous response, the work suggested by the reviewer has relied on a specific natural language processing approach - latent semantic analysis - and our work makes additional contributions beyond that approach by showing that more recent methods relying on embeddings from large language models, which include, in the new version of our manuscript, a fine-tuned model, can better predict psychometric characteristics of psychological measures relative to past approaches (e.g., Rosenbausch et al., 2020). In addition, we make further contributions of which we would like to highlight 1) a novel validation approach that considers structural fidelity (i.e., assignment of items to scales), 2) the validation of labels (testing different forms of embedding scales labels to increase match to scales), and 3) a relabeling method to eliminate jingle-jangle fallacies as a way to consolidate the scale-construct landscape. We now have rewritten parts of the introduction and discussion to, on the one hand, acknowledge past work and, on the other, make clearer the additional contributions of our work.

Specific feedback on evaluations:

1. Correlations between items from the same scale (Cronbach's Alpha): I'm not sure I've seen anyone else do this before, but it is a pretty obvious analysis that, due to its focus on items that measure (or belong to) the same underlying constructs, one would expect a high r . The relatively low correlations obtained (.28 - .53) are probably due to the focus on the personality domain, where scales were initially built differently than most other types of psychological theory.

We cannot provide a satisfying account for the absolute size of these correlations but agree that it is important to assess whether the methods we propose produce similar results across different types of scales/measures. We now make explicit the need to generalize results across different psychology domains in future validations in the Discussion.

2. Recovering the scales' structural fidelity, the authors consider a scale fully recovered when the z-score for each scale is above 2. This will require better justification and support from the literature.

To the best of our knowledge, there is no past empirical literature or meta-analytic evidence on values for structure fidelity. Our approach relies on adopting an arbitrary but reasonable criterion that is close to the observed structural fidelity obtained from empirical data and adopting that as a reference point. We have added a sentence in the results to clarify our use of this criterion and the comparison to empirical data that has now been significantly expanded by including additional data from four other data sets.

3. The authors employ another cutoff point without justifying it for automated jingle-jangle detection. The authors state as their cutoff point that a cosine difference of more than four times the average absolute difference between label and scale similarity and nearly three

times the average error in predicting empirical scale correlations represents a meaningful cutoff. Again, without justification, evaluation, or citations. It is worth mentioning that cosine scores are non-linear, which will affect the cutoff. I would recommend converting the cosines to a linear scale before introducing such a cutoff.

The reviewer makes a very important point that merits more attention than we provided in the previous version of our manuscript. We now address this issue in more detail by 1) acknowledging the difficulty in setting criteria, 2) reporting the robustness of results across different criteria (see supplemental materials), and 3) justifying the specific heuristic approach that we use to present the results for the specific cutoff values we adopt in our main analysis - that is, adopting a minimum and maximum number of hypothesized constructs. This approach only considers ordinal information and thus addresses potentially non-linearities. We hope that have made it clear that we do not argue for a specific criterion but, rather, suggest the adoption of arbitrary but reasonable ones that can be communicated and discussed by the research community by explicitly stating a minimum and maximum number of hypothesized constructs. That said, our robustness checks suggest that while the absolute value of candidate fallacies can vary considerably as a function of the criteria adopted, the relative proportion of jingle-jangle fallacies detected is similar for a given data set (e.g., IPIP), suggesting that this could be a more general characteristic of the conceptual and measurement landscapes considered. Moreover,

4. Also, for automated jingle-jangle detection, given the cutoff point the authors specify in my point #3, above, they then evaluate the success of the algorithm. This means that they only evaluate true positives vs. false positives, but miss the opportunity to evaluate true negatives vs. false negatives. They identify 60 jingle and 53 jangle fallacies, which is fewer jangle fallacies than I expected. Other research (such as Larsen and Bong) has shown that there are many times more jangle fallacies than jingle fallacies. It is possible this comes from the authors' more careful measurement of construct name similarity than that employed by Larsen and Bong, who used direct name match for this part of their evaluation. On the other hand, Larsen and Bong seem to have had access to a fully categorized set of constructs, which allowed them to address both type 1 and type 2 errors in their algorithms.

First, concerning the number of fallacies, as indicated in the previous point, the absolute number of fallacies is a direct consequence of the criteria used, consequently, this value can vary significantly depending on the criterion used. In turn, the relative number of jingle-jangle fallacies may be more interesting to consider as it is consistent across most criteria adopted.

Second, as indicated by the reviewer, one central difference between Larsen and Bong (2016) and our approach is that we adopt a similarity criterion rather than direct name matching. We believe this analytic choice can lead to a different pattern of results from Larsen and Bong (2016), but we would argue that our approach is appropriate because it helps handle cases that we know exist in the psychological literature of different spellings of the same construct (e.g., extraversion vs. extroversion), close synonyms (e.g., distrust vs mistrust), as well as more distant but problematic synonyms that represent overlapping constructs (e.g., grit vs self-control).

Third, concerning the possible advantage of Larsen and Bong (2016) relying on a categorized set of constructs, we would like to note that Larsen and Bong provide little evidence of expert agreement in their work and that we are not implying that the categorization provided by IPIP is superior or fully accepted by the community, only that it is widely used (the Goldberg et al. 2006 paper describing IPIP has been cited over 4000 times). We now make more explicit in the discussion that we think it important to conduct expert curation that builds on current efforts by the American Psychological Association and others that would allow a better foundation on which to conduct the type of efforts we propose in our work.

The solutions section of the paper is the one that most excites me. While cluster algorithms are notoriously unreliable and cosine similarities leave most of the variance unaccounted for (relative to supervised machine learning), this section is moving in a highly productive direction. I wish the authors had been more thorough in their literature review, as without it, it is hard to be sure of the actual contributions of the manuscript.

We hope we have addressed the reviewer's concern by providing a more explicit acknowledgment of past work adopting natural language processing methods to increase conceptual clarity. In turn, we appreciate the reviewer's acknowledgment that our work extends past work not only by showing that large language models seem to provide a more powerful tool to predict psychometric relations between measures but also that our proposed relabeling method can be used productively to produce more parsimonious taxonomies.

Minor typo: "and generated definitions form GPT-4."

Thank you for spotting this. We corrected the typo.

Reviewer #3:

The paper provides very insightful demonstrations of using LLM embeddings to identify (and even address) jingle-jangle fallacies in psychological measurements. I have only a few comments.

Thank you for the positive assessment and helpful feedback.

1) I think readers might need more guidance as to what they should do now. It appears that the paper—despite being very creative in the presented analyses—limits itself to be a demonstration for now. I wonder whether it can have more practical impact, maybe through policy suggestions for journals (e.g., maintenance of an openly searchable semantic space for scales), or tutorial materials, or an improved version of the online app in the Rosenbusch et al paper. I guess the latter would need users to have an openai api key but the costs for encoding a single scale are minimal.

We agree with the reviewer that practical applications such as an app or a tutorial could greatly benefit the community. Since the previous submission, we have written a tutorial paper that includes detailed information on how to use embeddings for similar purposes that we hope can be a step in this direction (see <https://doi.org/10.31234/osf.io/f7stn>). Further, we now include a yet more powerful fine-tuned model in our analysis and make this publicly available through a Hugging Face repository, thus facilitating the use of the methods we adopt in our work (<https://huggingface.co/dwulff/mpnet-personality>).

We also agree with the reviewer that further practical steps are crucial to improving the clarity of the field and have added a few sentences at the end of the discussion making clear that the idea of an openly searchable semantic space could be helpful to inform both expert discussion in general and journal review practices specifically.

2) The ipip data is great because it is easily accessible but they only constitute a very small part of the scale jungle in psychology. If the authors applied their approach to a larger corpus (e.g., the one of Rosenbusch et al.) that would not only lead to new insights but also allow for a sort of cross-validation of the decisions made for the ipip data. Plus it would offer a semantic space that scale developers and reviewers could query for existing scales.

In line with reviewer 3's suggestion, we have analyzed the Rosenbusch et al. (2020) data, which we now report in the supplementary materials. The results demonstrate how our approach can be generalized to a larger set of data, showing that jingle-jangle candidate fallacies can be identified and, potentially, resolved using our methodology. The main advantage of the Rosenbusch et al. (2020) data is its wide coverage of items and constructs; however, the curation of scale labels and items is less thorough than that for the International Personality Item Pool. As a result, we opted to report the IPIP analysis in the main text and the analysis of the Rosenbusch et al. (2020) data in the supplementary materials.

3) I would like to hear the authors' thoughts on the case when both scale content and label are highly similar. This is neither a jingle nor a jangle fallacy, but it is still bad because of

redundancy. So why not suggest cutting all scales with similar contents, regardless of their labels?

One main advantage of avoiding redundancy in measures is to increase clarity about which one to use to capture a specific construct. However, redundancy can be an asset when striving for reliability. Further, in intensive longitudinal or repeated-measurement designs, there can be advantages in not repeating the same items to avoid rote memorization. Given our uncertainty about the benefits of reducing redundancy, we have avoided proposing an overall method to decrease the number of scales available. However, we now expanded our discussion to be more explicit about this possibility as well as emphasizing the possibility of adopting an item-based approach to construct scales in a bottom-up fashion to obtain more coherent (and less redundant) measurements.

4) I think the term taxonomic incommensurability might not be the most accessible. I would suggest omitting or changing it for easy understandability and improved twitter marketing 😊

We understand that “taxonomic incommensurability” is not directly accessible to a lay audience. However, it is a technical term used in the epistemology literature (see Sankey, 1998) to refer to the phenomenon we are addressing. As a result, we hope that using the term helps reach a wider audience interested in the issue and can facilitate interdisciplinary dialogue to increase the reach of our contribution beyond psychology.

5) There were moments towards the beginning of the manuscript where I thought ‘labels’ referred to the words at the poles of item rating scales (so labels would be things like “strongly agree” or “not at all interested”)

We now refer to “construct labels” in the abstract and the first mentions of labels in the introduction to help dispel confusion.

6) I looked for an in-text link for supplementary materials and possibly pre-registration materials but I couldn’t find any.

Our work was not pre-registered but we provide a link to a repository including supplementary materials and code references in the “Code and data availability” section (<https://osf.io/nmv29/>).

Reviewer 1

The authors have done a good job of adding in more prior literature and comparisons. I think they have most of the content that they need, but I still find the article frustratingly hard to read. Terms are often used without definition until much later in the article. It is fine if some of the figures (especially in the supplement) are not important, but the key results are often hard to find.

Thank you for the overall positive evaluation. We have made several changes to improve clarity.

For example, I don't understand the authors' answer to my previous question "I don't know what the values range over. Figure 3D what are the units of "Scale-label alignment"? The text refers to cosine similarity but what does e.g. a "3" mean? Isn't this a similarity that can't exceed 1?" This is now panel F, and the legend says "Panel F illustrates our construct label evaluation. The points show the average alignment score as defined in the text ..." I believe you, but after a few minutes of searching in the text for "alignment" I gave up. Please put it where I can easily find it. For example when you first use the term.

We have added a definition of scale-label alignment in the method section. It quantifies the similarity between items in one scale to the corresponding label relative to the similarities between those items and other labels as a z-score. We also added an additional insertion in the main text:

In our validation strategy, we generated several types of label embeddings that we reasoned could have differential strengths and weaknesses and evaluated their relative alignment to scale content as operationalized by the scale embedding **using a z-score quantifying the similarity between scales and their associated label relative to all other labels.**

*Similarly, the article talks a lot about embeddings, but the introduction doesn't say what sort of embeddings they are. (Basically, a variation of RoBERTa, which one can guess from the 768 in the figure caption, or by going to the supplemental material, or by reading patiently until page 11, when you mention "Specifically, we fine-tuned MPNet [22]"--but don't say what MPNet is.) Please put a paragraph in the main text saying a little about what embedding you used and briefly explain **why** you used that embedding? Since you are computing similarities, I would have assumed that a model like sbert would have been better. Or that if you are fine-tuning, then maybe any Roberta-class model is pretty much equivalent?*

MPNet is one of the best-performing sentence-BERT models currently available. Similar to RoBERTa, it builds on the BERT architecture. We have selected the model because of its performance and ease of use for fine-tuning via the sentence-transformers library. The authors of the sentence-BERT (SBERT; https://sbert.net/docs/sentence_transformer/pretrained_models.html) project, Reimers and Gurevych, 2019 (which we now cite when justifying its use), also recommend it,

suggesting it is one of the best choices in this class. We clarify our choice of MPNet by adding the following two sentences to the main text.

This fine-tuned model is based on the MPNet model [20], which builds on the BERT architecture and is one of the best available sentence-BERT models [21]. The model was fine-tuned using 200 thousand pairs of personality items from several open datasets and is publicly available on Hugging Face (dwulff/mpnet-personality).

The supplement is equally hard to read. Can you state clearly (or maybe I just missed it) what is being done for fine-tuning? You are adjusting what (all the parameters in MPNet?) to reduce what loss function?

To clarify, we have added the following two sentences to the supplement:

As the loss function, we used the CosineSimilarityLoss function in the sentence_transformers library. This function computes the Euclidean distance between the cosines predicted by the model and the criterion, in this case, the empirical absolute Pearson correlations between personality items. We fine-tuned the entire model.

You concluded that your fine-tuned model is better than the other ones. That makes sense, but can you point me to (and cite in the main paper) the table that shows the numbers (e.g. correlations) for each of the embeddings that you used to reach that conclusion.

The comparisons between embedding models are spread over Figures S2-5. Figure S5 illustrates the crucial comparisons between the models' ability to predict the absolute correlations between item and scale scores. This analysis shows that the out-of-sample performance of the fine-tuned MPNet model (mpnet-ft-oos) is second only to the in-sample performance (mpnet-ft-is) and clearly outperforms all other models on most comparisons. We now refer to these Figures in the main text.

I'm also confused by some of the notation. E.g., "At the level of items, we observed extremely low mean absolute errors between $r = .026$ (FFM) and $r = .029$ (16PF) in-sample and higher but still low errors between $r = .07$ (BIG5) and $r = .065$ (HEXCO) outof-sample. These imply correlations of, on average, $r = .96$ (in-sample) and $r = .64$ (out-of-sample)." - I'm a bit confused by a "mean absolute error" being represented by the letter r , especially since the same paragraph uses r to be a correlation (which makes more sense to me). I think the MAE is something entirely different?

We apologize for this error and the confusion caused. We corrected the annotation and now use MAE to refer to performance measured in mean absolute error.

Finally, in the supplement, equation 2, what is the norm of n squared? You are doing a weighted sum in the numerator, weighting, but n_i , so would expect you would divide by $\|n\|$, and not square it.

The superscript two was intended to signify the use of the L2 norm. We have changed it to a subscript to avoid this confusion.

Reviewer 2

*As a staunch advocate of the necessity for research on taxonomic incommensurability, I firmly believe that a paper that effectively represents this research area not only deserves a place in *Nature: Human Behavior* but also contributes significantly to the advancement of our field. A paper must offer a unique contribution to be published in a top journal. In the initial version of the paper, the authors proposed an approach to address taxonomic incommensurability, suggesting it was the first to tackle the jingle-jangle fallacy using large language models.*

As should be apparent from the last round of review, the authors did not invent large language models and were not the first to use language models to address the jingle-jangle fallacy. The question is whether using a “large” language model rather than a “medium” language model represents novelty. Those of us who have worked in the area of language models for a few decades would probably agree that the computer scientists who developed innovations such as the attention mechanism (Vaswani et al. 2007) provided true novelty, but employing a large language model to an existing research area does not in itself represent novelty over existing work also employing language models to the same problem.

In short, we would likely not accept a paper that employed a new statistical approach (not developed by the authors themselves) to address self-efficacy without deeply engaging with the existing literature on self-efficacy (such as Bandura’s work). Given that the authors are not the first to address the jingle-jangle fallacy using language models, they must much more extensively engage with the relevant literature to explain the true novelty of their work over the existing work. Anything else would come across as academically dishonest and not in line with publication in a top journal. If the authors are unwilling to engage with the existing literature, this paper should not be published in NHB. I’m hoping the authors will be allowed to attempt this, though I’ve been discouraged after seeing two versions of their paper. As I have been requested by the editor of NHB to provide more information on the existing literature, I have tried to do that in the following.

We thank the reviewer for the detailed review and the additional challenge to clarify our contributions. We share the belief that work on taxonomic incommensurability remains of utmost relevance and agree with the importance of providing credit to past work while making clear what novel contributions (if any) a new manuscript provides. However, we disagree with the reviewer’s assessment that we failed to provide due credit to past work in this area or that our contributions are limited to applying a “large” language model to a problem that has otherwise been tackled.

We identify the following as our main contributions:

1. The conceptual introduction of the notion and empirical evidence of a jingle-jangle trade-off and highlighting that reducing one fallacy may increase the other - something that was not a focus of Larsen and Bong (2016), nor to our knowledge, any past contribution in this field.

2. The thorough evaluation and comparison of state-of-the-art **large** language models, including a fine-tuned model that we developed that outperforms all other existing models in several evaluations concerning the semantic relations between items, scales, and labels.
3. Consideration of the relatedness between construct labels and evaluation of several label types, advancing the character of jingle-jangle detection beyond past approaches that have focused only on the relation between measures and labels (e.g., Larsen & Bong, 2016)
4. Development of a rationale to decide on the thresholds used for jingle-jangle fallacies.
5. Proposal and demonstration of a relabeling approach to improve the mapping between measures and constructs while reducing fallacies and improving parsimony.
6. Generalization of the jingle-jangle tradeoff, detection, and reduction to a massive data set of a publicly available and often used set of personality scales (IPIP), as well as an additional dataset (Rosenbusch et al. 2022).

Obviously, our efforts to acknowledge past work were not perceived to be sufficient by the reviewer and we are happy to accommodate the reviewer's feedback to ensure that we provide a fair representation of the literature. As a result, we have rewritten portions of the introduction, specifically, we moved our paragraph referring to Larsen and Bong (2016) to earlier in the manuscript and make explicit that this work used language models to detect jingle-jangle fallacies. Simultaneously, we aimed to make clear that our contribution extends past work by considering not only the semantic relation between measures but also between constructs and well as a reduction of jingle-jangle fallacies through a new parsimonious taxonomy:

Past work has shown that natural language processing methods can be used to help clarify the relations between constructs and their measures [e.g., 11, 14, 15]. **Crucially, some approaches have been shown to identify jingle-jangle fallacies in an automated fashion across a large swathe of constructs [11].** However, these past efforts have utilized techniques, such as latent semantic analysis, that have now been superseded by more powerful language models that promise to capture even better several aspects of human psychology [e.g., 16–18]. A thorough comparison of different large language models, as well as an approach to detect jingle-jangle fallacies using these is lacking. **Moreover, past efforts have only considered gradual differences in the relatedness of measures but not fully the interconnections between constructs and their operationalizations, which is crucial for a full conceptual understanding of jingle-jangle fallacies, improving the mapping between measures and constructs, and reducing jingle-jangle fallacies through an improved taxonomy.**

The paper by Larsen and Bong (2016), published in the top journal of the Information Systems discipline, represented the first effort to address what they termed construct identity fallacies (a combination of the jingle fallacy and the jangle fallacy in a 2x2 framework) in behavioral and social sciences using natural language processing (NLP) techniques. Their work, particularly the Construct Identity Detector (CID) algorithm, which incorporates latent semantic analysis (LSA)

and improves it, sets a foundational precedent for detecting construct overlaps and differentiations. LSA was one of two vector space models examined and extended. As contemporary large language models (LLMs) like GPT-4 are scrutinized for their advancements, examining how these models are technically analogous to, yet an evolution of, the LSA-based methods described by Larsen and Bong is imperative. For example, given that Larsen and Bong (2016) calculated the ROC AUC's for their models, I think that is all you need to do for comparison and to prove that your models are better. It would be shocking if they weren't, but we would learn from knowing how much better without adding much extra effort.

Technical Parallels

- 1. Foundational Principles: LLMs and LSA are grounded in the vector space model. LSA reduces the dimensionality of text data to capture semantic meaning through singular value decomposition (SVD), a form of matrix factorization. Similarly, through transformer architectures, LLMs represent text in high-dimensional space using embeddings that encapsulate contextual semantics. In fact, it is often possible to treat LLM embeddings as LSA vectors and vice versa.*
- 2. Semantic Understanding: LSA captures word meanings based on their co-occurrence patterns in large corpora. LLMs extend this by training on massive datasets and using attention mechanisms to understand context over long text spans, improving semantic capture and context awareness.*
- 3. Application in Construct Identity: Larsen and Bong's CID1 combines LSA and other NLP algorithms to detect construct identities. LLMs can perform similar tasks more efficiently due to their training on diverse datasets, enabling them to recognize nuanced semantic relationships without extensive domain-specific tuning. This is worth mentioning as part of the justification for why your paper is worth publishing.*

Contributions of Larsen and Bong (2016)

- 1. Construct Identity Detector (CID): Larsen and Bong introduced CID, a tool that combines LSA with other algorithms to detect construct identity fallacies. This was a significant step towards addressing the jingle and jangle fallacies by providing an automated method to evaluate construct identities against human expert decisions.*
 - 2. Framework for Evaluating NLP Designs: They established a rigorous framework for comparing different NLP designs, demonstrating that hybrid models could outperform single-method approaches in construct detection tasks.*
 - 3. Implications for Behavioral Sciences: Their work highlighted the importance of addressing construct identity for the advancement of behavioral sciences, emphasizing the necessity for tools that can assist in literature reviews and meta-analyses by identifying construct overlaps. This is an area where a paper like yours would be expected to build further and carefully explain how the new model outperforms LSA and what this difference means for psychometric practice. It would benefit the authors to read the behavioral implications that came out of the Larsen and Bong paper and discuss whether their own work has additional implications.*
- While LLMs represent a significant technical advancement over traditional methods like LSA, the foundational principles remain similar. The authors do not claim to have built the original LLMs or contributed to their algorithmic development by inventing the attention mechanism, and they are extending an existing finding, the work of Larsen and Bong (2016) provides a crucial stepping stone for describing the authors' true contributions. Likely, their contribution lies in*

showing that they can now address construct identity fallacies more effectively. By engaging LLM capabilities, the authors have likely developed more accurate and efficient tools for construct identity detection, thereby contributing to enhancing the rigor and reliability of literature reviews and meta-analyses in social and behavioral sciences.

We recognize that a direct comparison between the Larsen and Bong (2016) approach, the *Construct Identity Detector (CID)* method was not included in our previous revision. This is because the approaches are not fully comparable (see details below) and we already included an extensive number of comparisons in our manuscript that suggest that an LSA-based approach would not provide a superior performance relative to the more recent embeddings.

In line with the reviewer's request, we now implemented the CID method with a few changes and report results of this comparison in the Supplementary Materials (section S6). We implemented the CID method with some modifications that we believe strengthened it relative to the implementation by Larsen and Bong (2016). First, instead of using LSA, we use the pretrained fastText word embedding model (<https://fasttext.cc/docs/en/english-vectors.html>) to determine word-word similarities. This model has been trained on hundreds of billions of tokens and has outperformed LSA in our analysis. Second, we use SUBTLEX-US (Brysbaert & New, 2009) instead of the Brown Corpus to determine word weights, because the former has been found to provide a better account of human evaluations.

We compared this implementation of the CID with other models concerning its ability to predict the correlation between items, which is the basis for being able to predict relationships between scales and between scales and labels. We found that the CID approach predicts the absolute correlations between items well. Specifically, we found correlations ranging between $r = 0.236$ (NEO) and $r = 0.350$ (FFM) and between MAE = 0.139 (FFM) and MAE = 0.158 (NEO). These values are higher than those observed for fastText alone or LSA, supporting the conclusion of Larsen and Bong that a combination of embeddings and other NLP approaches can be beneficial. However, these values are also far below those obtained for the large language models, including our fine-tuned model and other off-the-shelf language models. Specifically, our fine-tuned model achieved out-of-sample performance between $r = 0.546$ (NEO) and $r = 0.758$ (FFM) and between MAE = 0.065 (HEXACO) and MAE = 0.070 (BIG5), whereas the best off-the-shelf, the OpenAI large model achieved performances of $r = 0.440$ (NEO) and $r = 0.558$ (FFM) and between MAE = 0.118 (NEO) and MAE = 0.139 (FFM). This highlights the crucial differences between traditional word embedding models and recent text embedding models based on transformers, with the latter being better suited to evaluate the similarity of texts. We have added a new section in the Supplementary Materials (S6) that reports these results.

Finally, we did not conduct a comparison using ROC because our analysis does not involve a ground truth criterion (e.g., expert judgment) about the identity of

constructs/scales. Consequently, it is not possible to evaluate classification performance using methods such as ROC. That said, given that large language models account for empirical correlations between personality measures and constructs much more accurately, it is reasonable to expect that large language models would achieve a considerably higher classification performance were such judgments available. Crucially, it is unclear to what extent relying on expert judgments represents a productive approach given the current conceptual disarray in the literature. Unless expert judgments are obtained via a broad consensus process that can integrate the conflicting viewpoints present in modern (personality) psychology, there is the danger that any small-scale attempt at generating ground truth will produce only a piecemeal or inadequate representation of constructs and their relations. We now added qualifiers to our proposals concerning expert consensus to this effect in our discussion:

(...), our approach requires hypothesizing a minimum/maximum number of constructs needed in a specific research area, something that will likely be best established through **broad** expert consensus. In practice, this may amount to expert meetings **involving diverse perspectives or other consensus-building techniques** to discuss the utility of different constructs.

Response to reviewers

Review 2

From my point of view, the authors have done enough that I have no further negative comments. The only concerns I might raise at this point is that some of the descriptive language is still convoluted and probably not possible to follow for lay-people with little experience in language models or jingle jangle fallacies. A clear example is figure 1 and its description.

We have edited the annotation of Figure 1 to increase clarity.